# UNCOVERING THE WHY: INTERPRETABLE CLIP SIMILARITY VIA DUAL MODALITIES DECOMPOSITION

## ABSTRACT

The CLIP model has demonstrated strong capabilities in capturing the relationship between images and text through its learned high-dimensional representations. However, these dense features primarily express similarity via cosine distance, offering limited insight into the underlying causes of that similarity. Recent efforts have explored sparse decomposition techniques to extract semantically meaningful components from CLIP features as a form of interpretation. Nevertheless, we argue that these methods treat each modality independently, resulting in inconsistent decompositions that fail to reflect the cross-modal similarity from the aspect of concepts. In this paper, we introduce an explanation method for CLIP similarity via Dual Modalities Decomposition, CLIP-DMD, which employs a Sparse Autoencoder (SAE) to learn sparse decompositions of both CLIP image and text features within a shared concept space. To enhance interpretability, we propose two novel objectives: a Rate Constraint ($RC$) Loss, which promotes the crucial concepts to dominate the overall similarity, and a Corpus Cycle Consistency ($C^3$) Loss, which ensures that the most responsive features are both distinctive and accurately recognized by the encoder. To assess interpretability, we also design an evaluation protocol leveraging Large Language Models (LLMs) to provide automated and human-aligned assessments. Experimental results show that CLIP-DMD not only achieves competitive zero-shot classification, retrieval, and linear probing performance, but also delivers more human-understandable, reasonable, and preferable explanations of CLIP similarity compared to prior methods.

## 1 INTRODUCTION

Among the multimodal models which are developed to align information across different modalities, Contrastive Language-Image Pretraining (CLIP) (Radford et al., 2021) has demonstrated remarkable success in learning semantically rich representations of both image and text. By encoding inputs into a shared high-dimensional space, CLIP enables effective cross-modal alignment, achieving strong performance in tasks such as zero-shot classification and image-text retrieval. Typically, the similarity between image and text is measured using cosine similarity. However, this approach offers little insight into which components contribute to the perceived similarity. To address this gap, various methods have been proposed to uncover the underlying causes.

One line of work, inspired by the Concept Bottleneck Model (CBM) (Koh et al., 2020), seeks to map CLIP's image features to a human-understandable concept space. However, these approaches rely on annotated concept labels, which limits their scalability (Yuksekgonul et al., 2023; Selvaraj et al., 2024). To overcome this limitation, recent studies utilize CLIP's text encoder to expand the concept space using large text corpora (Rao et al., 2024; Shang et al., 2024; Tan et al., 2024). Nevertheless, these methods primarily use CLIP as an auxiliary tool to generate explanations for the image (i.e. the primary goal is to name the concepts found in the image embeddings), rather than interpreting the CLIP model itself.

In contrast, STAIR (Chen et al., 2023) retrains CLIP to align its features with an interpretable token space. More recent efforts (Gandelsman et al., 2024; 2025; Bhalla et al., 2024; Zaigrajew et al., 2025) explore CLIP's internal mechanisms or decompose its features into the combination of several concepts for the interpretability. Despite their contributions, these works focus mainly on the single modality, generally the image, lacking consideration of cross-modal similarity.

To bridge this gap, we propose CLIP-DMD, which explains CLIP similarity via Dual Modality Decomposition into a unified, interpretable concept space. Our approach provides both quantitative and human-understandable explanations for image–text similarity.

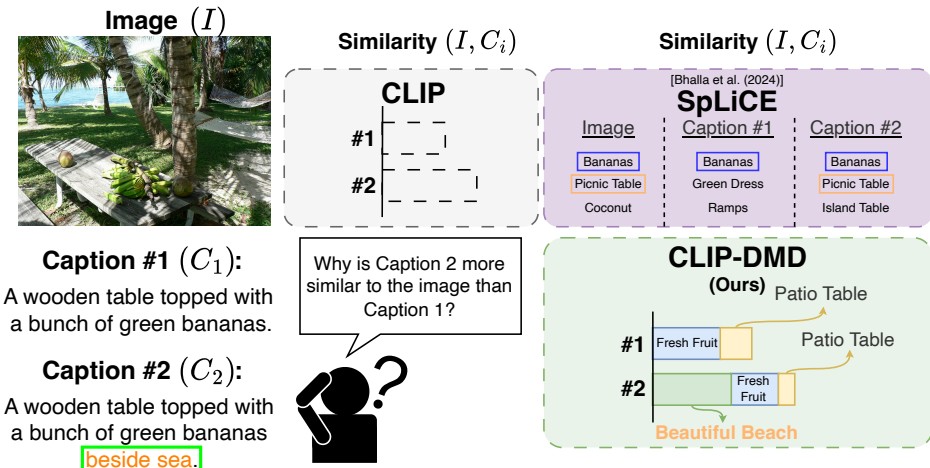

Figure 1: The "sea"-related concepts (i.e., the differences between the two captions highlighted with the green box and indicated in orange) are correctly identified by our CLIP-DMD and as the reason for the increase in the similarity between image and caption 2, but are missed by SpLiCE (Bhalla et al., 2024). In particular, SpLiCE incorrectly associates concepts such as "island table" and "picnic table", which do not align with the notion of being "beside the sea". In the top-mid example, the original CLIP model assigns a higher similarity to caption 2 than to caption 1, yet offers no explanation for this difference. In the top-right, SpLiCE identifies some shared concepts between the image and text, but fails to capture the conceptual distinctions between the two captions. The additional concepts unique to caption 2 are not properly reflected and do not accurately reflect proper similarity at the concept level. In contrast, the bottom-left side demonstrates that CLIP-DMD not only identifies the relevant shared concepts but also clarifies why caption 2 receives a higher similarity score than caption 1 at the concept level.

CLIP-DMD employs a Sparse Autoencoder (SAE) to map CLIP features into a higher-level sparse concept space. Unlike prior works that use separate decoders for each modality, we utilize a shared decoder to ensure that both image and text features are embedded in the same concept space. This shared decoder then learns the meaningful semantics, in which we call them learned corpus features, to reconstruct the CLIP feature with the encoded sparse features. Following Rao et al. (2024), we named these semantics after the training with the human-understandable concepts. To enhance the decoder's ability to recognize corpus features, we feed the learned concept weights from the decoder back into the encoder, encouraging accurate recognition. This process is formalized through the proposed Corpus Cycle Consistency ($C^3$) loss. Additionally, to enhance interpretability, we introduce another novel loss, Rate Constraint ($RC$) loss, making the similarity dominated by the critical concepts. Lastly, we incorporate an InfoNCE loss to train the sparse features to differentiate instances at the concept level, allowing the model to retain strong downstream performance. Our main contributions are as follows:

- We introduce CLIP-DMD, a novel framework that provides detailed, cross-modal explanations of CLIP similarity. Unlike prior works that either use CLIP as an auxiliary tool or focus solely on one modality, our method captures joint interpretability.

- We demonstrate that CLIP-DMD is a task-agnostic, robust, and efficient method, capable of providing meaningful insights into CLIP representations and highly related to the CLIP similarity without requiring full model retraining.

- Extensive experiments show that CLIP-DMD delivers competitive performance on downstream tasks while offering interpretable and concept-level explanations of CLIP's similarity.

## 2 RELATED WORKS

**Concept Bottleneck Models.** To enhance the interpretability of high-dimensional features, one common approach involves mapping these features into an intermediate concept space prior to the final classification layer. This concept space (often predefined and high-dimensional) serves as a bridge between abstract model representations and human-understandable semantics. A prominent example

Table 1: We compare our method against four representative categories of explainability approaches. CLIP-DMD offers concept-level decomposition across both image and text modalities, enabling interpretable insights without requiring an external concept corpus during training. Unlike methods that depend on task-specific classifiers to boost performance, CLIP-DMD remains task-agnostic by leveraging learned concept-level sparse features that are directly applicable to downstream tasks, achieving competitive performance while utilizing the linear classifier, as demonstrated in our further experiments (cf. Section 4).

| | Modality | Corpus-free Training | w/o Per-case Optimization | Cross-modal Similarity | Task Agnostic | w/o Classifier |
|---|---|---|---|---|---|---|
| PCBM (Yuksekgonul et al., 2023) | Image | ✗ | ✓ | ✗ | ✗ | ✗ |
| Res-CBM (Shang et al., 2024) | Image | ✗ | ✓ | ✗ | ✗ | ✗ |
| openCBM (Tan et al., 2024) | Image | ✓ | ✓ | ✗ | ✗ | ✗ |
| DN-CBM (Rao et al., 2024) | Image | ✓ | ✓ | ✗ | ✗ | ✗ |
| MSAE (Zaigrajew et al., 2025) | Image | ✓ | ✓ | ✗ | ✓ | ✗ |
| STAIR (Chen et al., 2023) | Image / Text | ✗ | ✓ | ✓ | ✓ | ✓ |
| SpLiCE (Bhalla et al., 2024) | Image / Text | ✗ | ✗ | ✓ | ✓ | ✓ |
| **CLIP-DMD (Ours)** | Image / Text | ✓ | ✓ | ✓ | ✓ | ✓ |

is the Concept Bottleneck Model (CBM) (Koh et al., 2020), which uses manually curated concepts as an intermediate layer. However, CBMs require substantial domain expertise and annotation effort, with its expressiveness being constrained by the fixed set of predefined concepts.

To alleviate such limitations, recent works (Yuksekgonul et al., 2023; Rao et al., 2024; Shang et al., 2024; Tan et al., 2024) utilize CLIP-based features to construct concept embeddings directly from text corpora, significantly reducing the annotation burden and enabling scalability to larger concept banks. For instance, PCBM (Yuksekgonul et al., 2023) leverages the CLIP text encoder to map textual concepts into the feature space, then assesses concept presence in images for classification via a learned predictor. DN-CBM (Rao et al., 2024) introduces a strategy to learn concepts without requiring predefined labels, allowing for more flexible and diverse semantic discovery, and then names the semantic feature after training. Res-CBM (Shang et al., 2024) analyzes components in CLIP image features and incrementally replaces them with the nearest concepts during training. OpenCBM (Tan et al., 2024) enables open-vocabulary interpretability by decomposing the classification head into interpretable components, allowing users to inject arbitrary concept sets into the model. In contrast to these methods, which generally treat CLIP features as auxiliary tools for bridging the gap to human understanding, our approach aims to directly interpret the semantics embedded within CLIP features. Rather than relying on external concept sets during training or fixed classifiers, we reveal what the CLIP features intrinsically represent through unified, modality-aligned sparse decomposition.

**Interpretable CLIP.** The impressive performance of CLIP features across a wide range of multimodal tasks has spurred growing interest in interpreting their high-dimensional representations. STAIR (Chen et al., 2023) introduces a token projection head that maps CLIP features into a sparse embedding space aligned with vocabulary tokens. While being effective, this approach requires training the model from scratch, limiting its transferability to other models. To better understand intermediate representations, TEXTSPAN (Gandelsman et al., 2024) and Gandelsman et al. (2025) analyze CLIP's internal mechanisms by identifying text tokens that characterize the functional roles of individual tokens. More recently, SpLiCE (Bhalla et al., 2024) adopts the linear representation hypothesis to express CLIP features as sparse linear combinations of interpretable CLIP text embeddings. Using LASSO regression, SpLiCE decomposes a CLIP representation into a weighted sum of text-derived basis vectors, offering a transparent view into the semantic structure of high-dimensional features. While these methods have advanced our understanding of CLIP's representations and modality-specific semantics, they primarily focus on interpreting individual modalities in isolation. As a result, they fall short in explaining the cross-modal similarity that is central to CLIP's design and use.

## 3 METHODS

We firstly present an overview of our proposed framework, CLIP-DMD, which utilizes a Sparse Autoencoder (SAE) to map CLIP features into a sparse, interpretable concept space as shown in Figure 2. Given CLIP image features $\mathcal{F}_{img}^{CLIP} \in \mathbb{R}^D$ and text features $\mathcal{F}_{text}^{CLIP} \in \mathbb{R}^D$, the modality-specific encoders, $Enc_{img}(\cdot)$ and $Enc_{text}(\cdot)$, project these features into a sparse representation: $\mathcal{F}_{img}^s = Enc_{img}(\mathcal{F}_{img}^{CLIP}) \in \mathbb{R}^M$ and $\mathcal{F}_{text}^s = Enc_{text}(\mathcal{F}_{text}^{CLIP}) \in \mathbb{R}^M$, where $M$ denotes the sparse dimension. These sparse features capture the conceptual composition of the original CLIP representations, offering human-interpretable semantics.

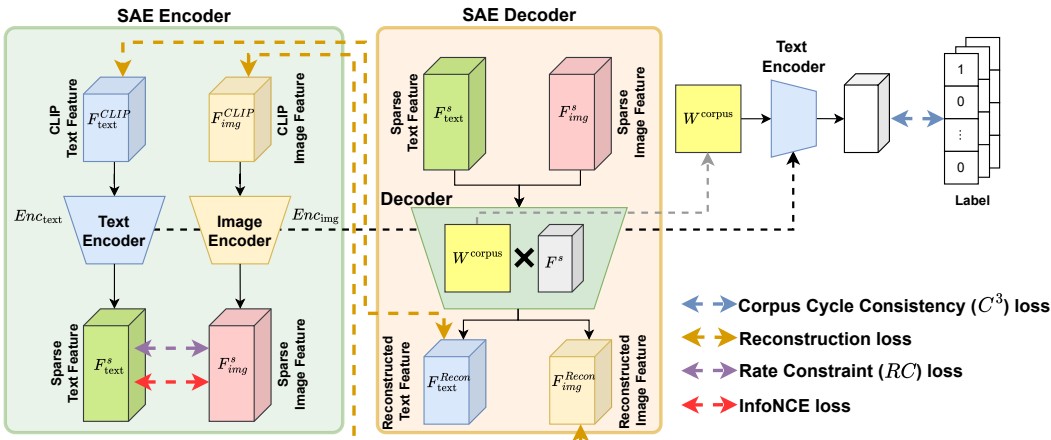

Figure 2: **CLIP-DMD training workflow.** We begin by extracting sparse concept features from CLIP image and text embeddings using their respective encoders. These sparse features are supervised using both Rate Constraint ($RC$) and InfoNCE losses to enforce interpretability and discriminability. The sparse features are subsequently passed through a shared decoder, which reconstructs the original CLIP features using a set of learnable weights representing meaningful concepts (yellow box). To ensure faithful reconstruction, we apply a Reconstruction loss that encourages the decoded features to closely match the original CLIP ones. Finally, to strengthen the encoder's ability to recognize and respond to concept embeddings, we apply the Corpus Cycle Consistency ($C^3$) loss by feeding the learnable concept weights into the decoder and supervising their corresponding sparse activations.

To ensure that both image and text features are aligned within a unified concept space, we employ a shared decoder $Dec(\cdot)$ across modalities. The decoder reconstructs the original CLIP features from the sparse representations, yielding $\mathcal{F}_{img}^{Recon} = Dec(\mathcal{F}_{img}^s) \in \mathbb{R}^D$ and $\mathcal{F}_{text}^{Recon} = Dec(\mathcal{F}_{text}^s) \in \mathbb{R}^D$. This design facilitates cross-modal alignment and departs from previous approaches that often interpret only a single modality in isolation without considering the relation across modalities. Furthermore, we incorporate the InfoNCE loss (Oord et al., 2018) to train the sparse features, encouraging them to retain discriminative power for downstream tasks while enhancing interpretability. Unlike prior methods that trade interpretability for performance, our approach strikes a balance by achieving both. In the following subsections, we detail the key components and design choices of CLIP-DMD.

### 3.1 CORPUS CYCLE CONSISTENCY ($C^3$) LOSS

Given a CLIP feature $\mathcal{F}_m^{CLIP} \in \mathbb{R}^D$, where $m \in [img, text]$, the corresponding encoder (i.e. $Enc_{img}$ or $Enc_{text}$) is trained to produce a sparse representation $\mathcal{F}^s$ that captures the contribution of each underlying concept present in the CLIP feature. Using these sparse features, the decoder reconstructs the original CLIP feature via a set of learnable basis vectors, denoted as $W^{corpus}$, which we refer to as the learnable corpus features. That is, $\mathcal{F}_m^{Recon} = Dec(\mathcal{F}_m^s) = W^{corpus} \cdot \mathcal{F}_m^s$ where $m \in [img, text]$. Intuitively, the encoder should be capable of correctly recognizing and activating these corpus features to reflect the key conceptual factors embedded in the CLIP feature. To encourage this behavior, we introduce the Corpus Cycle Consistency ($C^3$) Loss, which promotes the encoder's ability to identify and respond accurately to the learned corpus features.

Specifically, when a corpus feature $w_i \in W^{corpus}$ is passed through the text encoder $Enc_{text}$, the resultant sparse feature $\mathcal{F}_i^{corpus} = Enc_{text}(w_i)$ should exhibit the highest activation at the corresponding $i$-th dimension. This ensures that the encoder maintains semantic fidelity in recognizing meaningful components. The $C^3$ loss is formally defined as:

$$\mathcal{L}^{C^3}(\mathcal{F}_i^{corpus}) = \mathbf{CE}(\mathcal{F}_i^{corpus}, \mathbf{1}_i),\qquad(1)$$

where $\mathbf{CE}$ denotes cross-entropy, and $\mathbf{1}_i$ is a binary vector with $i$-th element equals to 1 and 0 elsewhere.

### 3.2 RATE CONSTRAINT (RC) LOSS

Given a pair of CLIP features (i.e. an image feature $\mathcal{F}_{img}^{CLIP}$ and a text feature $\mathcal{F}_{text}^{CLIP}$), their similarity is typically computed using cosine similarity $\mathbf{Cos}$:

$$\mathbf{S}(\mathcal{F}_{img}^{CLIP}, \mathcal{F}_{text}^{CLIP}) = \mathbf{Cos}(\mathcal{F}_{img}^{CLIP}, \mathcal{F}_{text}^{CLIP}).\qquad(2)$$

While this metric quantifies how similar two features are, it does not reveal why they are similar, i.e. which concepts contribute to the similarity.

To address this limitation, we propose an interpretable similarity computation based on the presence of shared concepts. Specifically, the sparse features produced by our encoders (i.e. $Enc_{img}$ and $Enc_{text}$) should capture the concept-wise composition of the image and text inputs (i.e. each sparse feature encodes the activation weights over a learned concept basis $W^{corpus}$, in which we could later adopt the sparse features and $W^{corpus}$ to perform reconstruction of CLIP features). Using these sparse features, we redefine similarity in a concept-aware manner, using the same cosine metric:

$$\mathbf{S}(\mathcal{F}_{img}^s, \mathcal{F}_{text}^s) = \mathbf{Cos}(\mathcal{F}_{img}^s, \mathcal{F}_{text}^s). \tag{3}$$

This formulation ensures that similarity arises from shared and interpretable concepts (i.e. those activated simultaneously in both image and text) thereby offering a transparent rationale for the observed alignment.

To generate interpretable explanations from the sparse features, it is essential to highlight a small number of significantly contributing concepts. Ideally, these top contributing concepts should dominate the overall similarity, thereby offering a concise and reliable interpretation. To encourage this property, we propose the Rate Constraint ($RC$) Loss, which maximizes the contribution ratio of the top-$\mathbf{k}$ most salient concepts relative to all others. Building on the similarity formulation in Equation 3, we redefine the similarity based on the top-$\mathbf{k}$ contributing concepts as follows:

$$\mathbf{S^k}(\mathcal{F}_{img}^s, \mathcal{F}_{text}^s) = \sum \mathbf{sort^k}(\tilde{\mathcal{F}}_{img}^s \odot \tilde{\mathcal{F}}_{text}^s), \tag{4}$$

where $\tilde{\mathcal{F}}$ denotes the $\mathcal{F}$ feature normalized to unit length, $\odot$ represents the element-wise product, and $\mathbf{sort^k}$ selects the top-$\mathbf{k}$ largest values. This formulation emphasizes the shared high-contribution concepts between modalities. The Rate Constraint Loss is then defined as:

$$\mathcal{L}^{\mathbf{RC}}(\mathcal{F}_{img}^s, \mathcal{F}_{text}^s) = 1 - \mathbf{S}^k(\mathcal{F}_{img}^s, \mathcal{F}_{text}^s)/\mathbf{S}(\mathcal{F}_{img}^s, \mathcal{F}_{text}^s), \tag{5}$$

which penalizes cases where top-$\mathbf{k}$ concepts do not account for a substantial portion of the similarity.

The overall training objective combines the reconstruction, interpretability (aforementioned $C^3$ and $RC$ losses), and the typical contrastive components:

$$\sum_{m \in (img, text)} \mathcal{L}^{MSE}(\mathcal{F}_m^{Recon}, \mathcal{F}_m^{CLIP}) + \mathcal{L}^{C^3}(\mathcal{F}^{corpus}) + \mathcal{L}^{RC}(\mathcal{F}_{img}^s, \mathcal{F}_{text}^s) + \mathcal{L}^{IN}(\mathcal{F}_{img}^s, \mathcal{F}_{text}^s),$$

$$\tag{6}$$

where $\mathcal{L}^{MSE}$ denotes the mean squared error loss for reconstruction, and $\mathcal{L}^{IN}$ is the typical InfoNCE loss which pulls paired image and text representations closer in the sparse concept space.

### 3.3 LLM Evaluation

Evaluating the quality of generated explanations remains a challenging and open problem. Prior works (Fel et al., 2023; Rao et al., 2024) have relied on human-subject studies, typically through questionnaires, to assess the interpretability of model outputs. While being informative, these approaches are labor-intensive and do not scale well to large datasets.

To address this limitation, we introduce an LLM-based evaluation framework that enables scalable comparison across different interpretability methods. Given a set of inputs, including explanations from two methods (the list of concepts shared between the image and caption, along with the corresponding sparse-feature similarity produced by each method), the LLM is tasked with assessing which explanation provides more insightful reasoning for the observed similarity. To mitigate positional bias (e.g., a tendency to favor the first option), we instruct the LLM to score each method's interpretation independently rather than choose between them. We additionally randomize the order in which the two methods are presented to reduce ordering effects. After scoring all samples using this procedure, we compute how many times each method receives the highest score from the LLM to determine their relative interpretability. Additionally, to improve robustness and reduce the bias associated with relying on a single LLM, we also introduce an ensemble LLM evaluation criterion. This approach leverages multiple LLMs to assess the interpretations and collectively determine the final judgment. Each LLM serves as an independent expert, evaluating the provided explanations from different perspectives, and the final preference is based on majority agreement among these

Table 2: The zero-shot performance evaluation on CUB and ImageNet, and text-to-image and image-to-text retrieval performance on MS-COCO benchmarks with different backbone choices. The CLIP presents the original CLIP model without providing any interpretability capability. The reported values represent the mean performance ($\pm$ 1 standard deviation).

| Backbone | Method | interpret-ability | CUB | | ImageNet | | MS-COCO | | | |
|---|---|---|---|---|---|---|---|---|---|---|
| | | | R@1 | R@5 | R@1 | R@5 | T2I R@1 | T2I R@5 | I2T R@1 | I2T R@5 |
| ResNet50 | CLIP | | 46.69 | 81.14 | 57.94 | 84.82 | 28.29 | 52.95 | 48.16 | 73.90 |
| | SpLiCE | ✓ | 6.40 | 23.51 | 30.99 | 56.14 | 5.60 | 16.33 | 8.62 | 22.26 |
| | Ours | ✓ | $34.25_{\pm0.52}$ | $69.20_{\pm0.66}$ | $53.03_{\pm0.13}$ | $82.50_{\pm0.15}$ | $31.90_{\pm0.16}$ | $56.90_{\pm0.07}$ | $45.15_{\pm0.38}$ | $71.65_{\pm0.28}$ |
| ViT-B-32 | CLIP | | 65.50 | 90.90 | 66.30 | 89.45 | 38.79 | 64.79 | 56.58 | 80.26 |
| | SpLiCE | ✓ | 7.34 | 26.48 | 34.95 | 60.90 | 6.04 | 16.59 | 10.60 | 25.46 |
| | Ours | ✓ | $54.03_{\pm0.40}$ | $85.49_{\pm0.32}$ | $61.98_{\pm0.17}$ | $87.51_{\pm0.04}$ | $36.64_{\pm0.15}$ | $62.81_{\pm0.08}$ | $51.25_{\pm0.14}$ | $76.35_{\pm0.11}$ |
| ViT-B-16 | CLIP | | 70.50 | 92.99 | 69.38 | 91.19 | 41.65 | 67.16 | 59.30 | 82.42 |
| | SpLiCE | ✓ | 7.59 | 30.72 | 38.34 | 65.08 | 6.66 | 18.11 | 11.80 | 26.98 |
| | Ours | ✓ | $57.79_{\pm0.69}$ | $88.29_{\pm0.37}$ | $64.80_{\pm0.14}$ | $89.22_{\pm0.00}$ | $39.43_{\pm0.09}$ | $65.44_{\pm0.08}$ | $54.05_{\pm0.27}$ | $78.08_{\pm0.21}$ |

experts. Following the same procedure used in the single-LLM evaluation, we ask various LLM models to score the interpretations produced by each method across the dataset and record how often each method receives the most votes. Based on these aggregated counts, we determine which method provides the most preferred and consistently favored interpretation.

## 4 EXPERIMENTS

**Settings.** We utilize the CLIP model (Radford et al., 2021) with a ViT-B/32 vision encoder and its corresponding text encoder to extract image and text features from the CC3M (Sharma et al., 2018) dataset. These pre-extracted features are then used to train our Sparse Autoencoder (SAE) for 32 epochs, using the AdamW optimizer with a learning rate of 5e-4, a weight decay of 1e-4. The image and text encoders, along with the shared decoders, each consist of a single fully connected layer without a bias term. The encoder outputs are followed by a ReLU activation to prevent negative concept weights. We set $k$=32 (for the $RC$ loss) and $M$=15360 (the dimensionality of sparse features). To enhance the diversity of image features during training, Gaussian noise with a standard deviation of 0.03 is added as data augmentation. To interpret the semantic concepts learned by the decoder, we follow SpLiCE (Bhalla et al., 2024) to construct a corpus from the most frequent unigrams and bigrams in the LAION-400M (Schuhmann et al., 2021) dataset captions. All experiments are trained on 4 GPUs of RTX 3090.

**Datasets.** Four datasets are adopted for our evaluation:
○ **CUB_200_2011** (Wah et al., 2011) is a fine-grained image classification dataset containing 11788 images across 200 bird species, with 5994 images for training and 5794 for testing. In our experiments, we only use the validation images and their species labels to test zero-shot performance.
○ **MS-COCO** (Lin et al., 2014) is a large-scale benchmark dataset for image recognition, segmentation, and captioning, comprising over 330000 images, more than 200000 of which are labeled across 80 object categories with instance masks, object labels, and five human-written captions per image. For our evaluation, we use the validation set containing 5000 images with corresponding captions to assess text-to-image and image-to-text retrieval performance.
○ **ImageNet-1K** (Russakovsky et al., 2015) is a large-scale dataset for image classification and object recognition, consisting of around 1.2 million images across 1000 classes. In our experiments, we use the 50000 image validation set to evaluate zero-shot performance across all classes and assess the linear probing with training on the training set and evaluation on the validation set.
○ **Tiny ImageNet** (mnmoustafa & Ali, 2017) is a widely used benchmark derived from the original ImageNet dataset, specifically tailored for efficient evaluation of image classification models under limited computational resources. It comprises 200 object categories, with each class containing 500 training images, 50 validation images, and 50 test images. We evaluate the linear probing by training on the training set and evaluating on the validation set.
We provide additional experimental results on more datasets in the supplementary material.

### 4.1 QUANTITATIVE RESULTS

Here, we show that the sparse features learned by CLIP-DMD achieve competitive performance compared to existing interpretability methods, particularly in capturing cross-modal relationships that lead to improved results. For these learned sparse features, which represent concept compositions and serve as a bridge between CLIP features and human recognition by revealing the composition of

Table 3: Linear probing evaluation conducted on CUB and TinyImageNet datasets, via learning a linear classifier to classify the extracted sparse concept features from images.

| Backbone | Method | CUB | TinyImageNet |
|---|---|---|---|
| ResNet50 | CLIP (Radford et al., 2021) | $44.40_{\pm1.25}$ | $53.51_{\pm0.05}$ |
| | SpLiCE (Bhalla et al., 2024) | $9.61_{\pm0.38}$ | $41.60_{\pm0.10}$ |
| | DN-CBM (Rao et al., 2024) | $51.44_{\pm0.83}$ | $49.23_{\pm0.02}$ |
| | **CLIP-DMD (Ours)** | $48.80_{\pm0.15}$ | $55.75_{\pm0.04}$ |
| ViT-B-32 | CLIP (Radford et al., 2021) | $53.88_{\pm0.31}$ | $75.19_{\pm0.06}$ |
| | SpLiCE (Bhalla et al., 2024) | $13.78_{\pm0.91}$ | $61.68_{\pm0.12}$ |
| | DN-CBM (Rao et al., 2024) | $69.15_{\pm0.18}$ | $74.96_{\pm0.06}$ |
| | **CLIP-DMD (Ours)** | $72.12_{\pm0.12}$ | $75.35_{\pm0.02}$ |
| ViT-B-16 | CLIP (Radford et al., 2021) | $62.31_{\pm0.34}$ | $77.77_{\pm0.06}$ |
| | SpLiCE (Bhalla et al., 2024) | $13.95_{\pm1.01}$ | $65.11_{\pm0.04}$ |
| | DN-CBM (Rao et al., 2024) | $76.35_{\pm0.34}$ | $77.53_{\pm0.04}$ |
| | **CLIP-DMD (Ours)** | $78.12_{\pm0.05}$ | $77.47_{\pm0.04}$ |

Table 4: Spearman's rank correlation coefficient to measure how each method reflects distribution of CLIP similarity.

| | MS-COCO |
|---|---|
| SpLiCE (Bhalla et al., 2024) | 0.0793 |
| CLIP-DMD (Ours) | 0.4339 |

Table 5: Latency (seconds per sample) of inference time.

| | Latency |
|---|---|
| SpLiCE (Bhalla et al., 2024) | 0.38 |
| DN-CBM (Rao et al., 2024) | 0.008 |
| CLIP-DMD (Ours) | 0.002 |

concepts in high-dimensional space, we would like to evaluate whether these concept-level features can be used for downstream tasks. Specifically, for CLIP-DMD and SpLiCE, we utilize each method to extract sparse features that represent the concept composition of the CLIP features, and evaluate them on zero-shot classification and retrieval tasks to assess whether the sparse features, presenting the factor of concept, can effectively reflect the similarity between images and texts. This examines the capability of reflecting the similarity across modalities by using the identified concept. As shown in Table 2, CLIP-DMD outperforms the interpretable method SpLiCE (Bhalla et al., 2024) on zero-shot classification and image-text retrieval tasks, which SpLiCE might lack the capability to present the concept in a consistent manner across modalities, leading to severe degradation. Though CLIP-DMD achieves worse performance than the baseline, it provides concept-level insight with its sparse features, which CLIP lacks in relative terms. These results indicate that our method effectively projects CLIP features into a consistent concept space suitable for cross-modal alignment, useful for presenting the similarity at the concept level. (See the supplementary materials for additional comparisons across various datasets.)

Prior methods (Yuksekgonul et al., 2023; Rao et al., 2024; Shang et al., 2024; Tan et al., 2024) decompose image and text features independently, producing inconsistent concept spaces that prevent zero-shot and retrieval evaluation. SpLiCE resolves this by using a large predefined corpus to align concept spaces, but it still relies on modality-specific decomposition and requires re-optimization for each case, leading to weak similarity reflection when using sparse, concept-level features. In our evaluation, we apply SpLiCE to its decomposed sparse features to measure how well its concept-based similarity aligns, this differs from its original design. In contrast, CLIP-DMD uses a shared decoder to ensure both modalities operate in the same concept space, and employs InfoNCE loss to pull paired sparse features closer together while separating non-paired ones, effectively addressing these limitations.

Next, we further assess whether the learned sparse features are discriminatively while presenting the concept compositions. We conduct a linear probing experiment in which the extracted sparse features are used to classify images by training a single fully connected layer for 100 epochs, optimized with AdamW, learning rate starting with $10^{-4}$ for every dataset. As shown in Table 3, our learned sparse features perform competitively on linear probing tasks across multiple backbones and datasets. This demonstrates that CLIP-DMD can learn sparse features with a discriminative representation while presenting the composition of the concept carried in the CLIP features, resulting in competitive performance to other interpretable methods and par with or surpassing the baseline. Moreover, we evaluate the trend of similarity between CLIP and other methods to determine whether each method can accurately reflect the similarity behavior of CLIP. As shown in Table 4, we use Spearman's rank correlation coefficient to assess the alignment between the similarity trends produced by our method and SpLiCE with those of CLIP on the MS-COCO. This is done by comparing the ranked similarity scores of captions associated with the same image (e.g., comparing the similarity sequence order calculated by CLIP and compared methods). The results show that CLIP-DMD has a higher correlation with the similarity trend of CLIP than SpLiCE, meaning our method can provide more accurate reasons for the difference in CLIP similarity while also reflecting the similarity. Finally, Table 5 reports inference times. CLIP-DMD is more efficient due to its simpler design, whereas SpLiCE requires per-sample re-optimization with LASSO, leading to significantly higher latency.

## 4.2 QUALITATIVE RESULTS

**Task Agnosticity of Concepts.** Here, we provide samples that demonstrate CLIP-DMD learns task-agnostic, robust, and semantically meaningful concepts from the CC3M dataset, as well as present its robustness on both the ImageNet and MS-COCO datasets. As illustrated in Figure 3, we present the top-4 most responsive images for each discovered concept, with the high similarity between the learned corpus features and predefined corpus features extracted by CLIP. The samples demonstrate that our method captures semantically coherent features that align well with their automatically assigned concept names. Even in the various datasets, the learned corpus feature consistently reflects the correct and correlated meaning of concepts. Additionally, the learned concepts exhibit a wide range of complexity, from simple object-level categories (e.g., Toddlers, Dinner Table) to more abstract or scene-level concepts (e.g., Grass Field, Elegant Wedding), highlighting the expressiveness and versatility of the learned concept space.

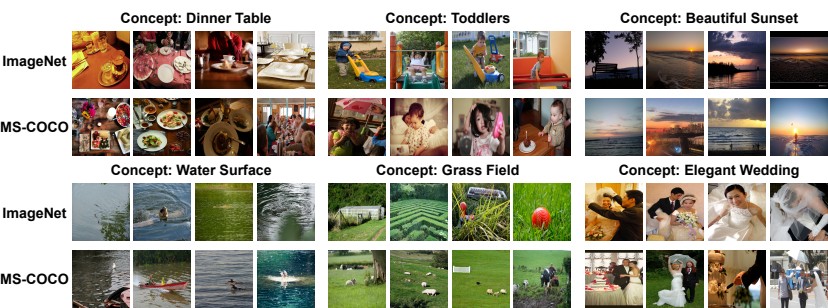

Figure 3: The sampled concepts demonstrate that the features learned by CLIP-DMD consistently and accurately capture the corresponding semantic concepts.

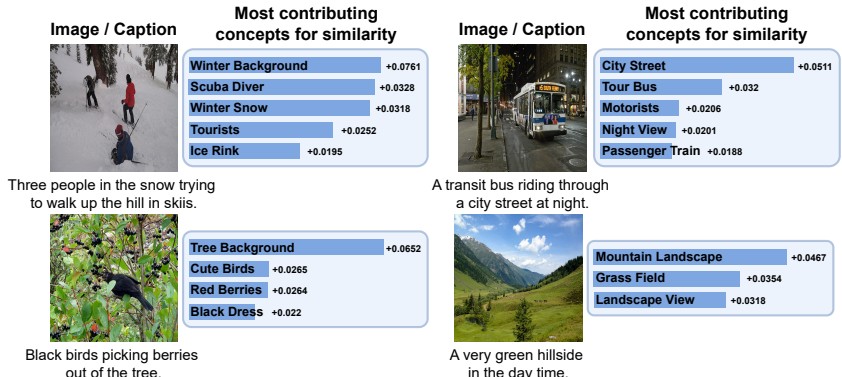

Figure 4: We present explanations generated by CLIP-DMD to illustrate why a given image and caption are considered similar. CLIP-DMD decomposes both image and text CLIP features into concepts and identifies the most contributing ones as the rationale behind the similarity.

**CLIP-DMD Explanations.** We show that CLIP-DMD can provide deeper and more interpretable insight into CLIP similarity. For each instance, we begin by decomposing both image and text features into a set of sparse concept representations. These sparse features are then used to compute similarity scores, from which we extract the top-$k$ most contributing concepts (cf. Section 3.2). These selected concepts highlight the shared semantic content between the image and text, and predominantly drive the similarity score. We present some samples in Figure 4, CLIP-DMD effectively identifies shared concepts across both modalities, providing a clear and interpretable rationale for the resulting CLIP similarity from the concept level. Beyond this, CLIP-DMD is also capable of explaining differences in similarity scores. In Figure 5, the second set of captions often omits key details that are visually present, whereas the first set captures these elements more comprehensively. Our method captures this distinction by revealing additional concepts in the first explanations, which account for their higher similarity scores relative to the second. In contrast, SpLiCE fails to highlight such semantic discrepancies, offering limited explanatory power for variations in CLIP similarity.

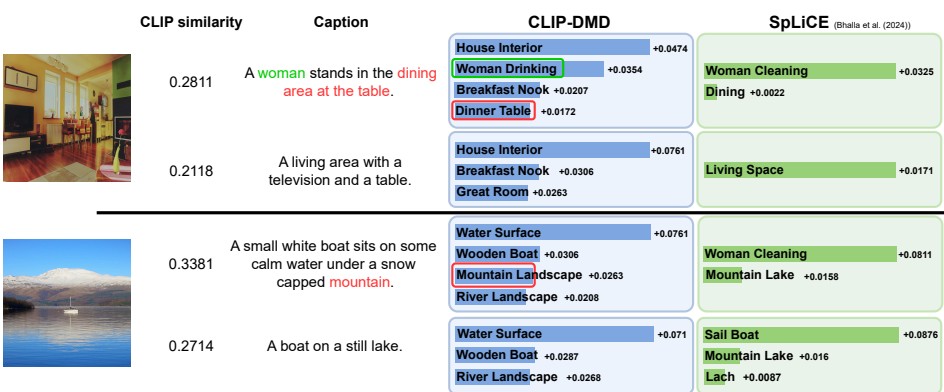

Figure 5: Examples show that CLIP-DMD effectively accounts for disparities in CLIP similarity, while SpLiCE fails to offer meaningful explanations. In each case, CLIP-DMD accurately identifies key informational differences and reflects them in the generated explanations, e.g highlighting "woman" and "dinner table" in the first example, and "mountain" in the second. In contrast, SpLiCE does not capture or reflect these key elements, leading to less informative or incomplete explanations.

Table 6: The interpretability comparison between CLIP-DMD and SpLiCE on the MS-COCO dataset, along with the human-alignment rate of the ensembling evaluated model. The ensemble model places more emphasis on CLIP-DMD than SpLiCE and achieves a suitable alignment rate with humans.

| Evaluated Model | Interpretation Comparison | | Human Alignment Rate |
| | CLIP-DMD (Ours) | SpLiCE | |
|---|---|---|---|
| LLaMA-3.2-11B-Vision-Instruct + Gemma-3-27B-it + Qwen2.5-VL-32B-Instruct | $64.16 \pm 0.13$ | $35.84 \pm 0.13$ | $72.00 \pm 2.00$ |

*The experiment is conducted three times, and the results are reported as the mean and standard deviation to account for randomness introduced by permutation and ordering.

## 4.3 INTERPRETABILITY EVALUATION

To assess interpretability from a human preference perspective, we adopt the evaluation method described in Section 3.3, comparing the explanations generated by CLIP-DMD and SpLiCE. The evaluations utilize the ensemble models with LLaMA-3.2-Vision-11B-Instruction (Grattafiori et al., 2024), Gemma-3-27B-it (Team et al., 2025), and Qwen2.5-VL-32B-Instruct (Bai et al., 2025) to improve the robustness and reduce the bias associated with relying on a single LLM. The experiment is conducted on the MS-COCO dataset, using the first caption associated with each image as the reference text. The result in Table 6 indicates that CLIP-DMD provides superior interpretability compared to SpLiCE, which is decided by integrating the decisions made by each LLM model and summarizing the preference. SpLiCE decomposes each modality independently, which often results in noisy and inconsistent concepts between the image and text branches. As a consequence, only a small subset of highly precise and strongly aligned concepts overlap, causing low-relevance but meaningful semantic cues to be omitted. Moreover, because the decompositions are performed separately, semantically similar concepts may be mismatched across modalities, leading to further misalignment in the presented explanations. These limitations also contribute to weaker correlation with the original CLIP similarity trends (cf. Table 4) and reduced performance on downstream tasks (cf. Table 2). In contrast, CLIP-DMD jointly decomposes image and text features into a shared concept space, resulting in more coherent explanations. Importantly, this interpretability improvement is achieved without significantly compromising downstream performance, thanks to the objective design (e.g., the InfoNCE loss) that preserves the discriminative power of the learned sparse features. To further validate the reliability of LLM-based evaluations, we conduct a human alignment study for each evaluated model. We randomly sample 50 cases from the dataset, present humans with the same information used in the LLM prompts, and ask them to choose the more appropriate interpretation. By comparing human selections with LLM decisions, we measure how closely each model aligns with human judgment. As shown in Table 6, the ensemble of LLM models exhibits a suitable alignment with human choices, reinforcing the reliability of their assessments. We also provide the single model result for interpretability comparison and the human alignment rate for the single model in the supplementary material.

Table 7: Interpretability comparison between ablation variants. The values in each column represent the percentage of samples in the dataset for which each method's explanation is preferred by the LLM evaluator in terms of clarifying the similarity between image and text. (Evaluated using LLaMA three times, and we report the mean and standard deviation.)

| $C^3$ loss | $RC$ loss | Interpretability | | |
|:---:|:---:|:---:|:---:|:---:|
| ✓ | ✓ | $78.02\%_{\pm 0.23\%}$ | $50.29\%_{\pm 1.03\%}$ | $79.29\%_{\pm 0.36\%}$ |
| ✓ | - | $21.98\%_{\pm 0.23}$ | - | - |
| - | ✓ | - | $49.71\%_{\pm 1.03\%}$ | - |
| - | - | - | - | $20.71\%_{\pm 0.36\%}$ |

**Image**

| w/ RC loss and $C^3$ loss | w/ RC loss | w/ $C^3$ loss | None |
|---|---|---|---|
| Chocolate Brown +0.0525 | Cake Recipe +0.1093 | Festival +0.0043 | Basketball Player +0.0027 |
| Cake Recipe +0.0511 | Chocolate Cake +0.0422 | Wedding Cake +0.0026 | Ideas Para +0.0024 |
| Food Ideas +0.0309 | | Baby Gift +0.0024 | Title Screen +0.0022 |

**Caption:**
A piece of chocolate cake on a plate.

Figure 6: Qualitative analysis of the proposed $C^3$ and $RC$ losses. Without either loss, the generated explanations lack reliability and coherence. The RC loss encourages similarity to depend more heavily on key concepts, enhancing interpretability. The $C^3$ loss strengthens the encoder's ability to recognize concepts, an effect that may not be visually apparent in qualitative examples but is reflected in improved quantitative performance (cf. Table 7). When both losses are combined, the explanations become more accurate and conceptually meaningful.

### 4.4 ABLATION STUDY

Here we conduct ablation study for the proposed Corpus Cycle Consistency ($C^3$) and Rate Constraint ($RC$) losses, evaluating both performance and interpretability against the full CLIP-DMD model. We first examine performance results, as presented in Table 8. The model trained with only the $C^3$ loss achieves the highest overall performance, driven by improved recognition of meaningful concepts from CLIP features. In contrast, using only the $RC$ loss results in a slight drop in performance, as it enforces similarity to rely heavily on a few dominant concepts, potentially overlooking other useful information. Furthermore, we evaluate interpretability using both quantitative and qualitative analyses (Table 7 and Figure 6, respectively). Removing both losses leads to a significant decline in interpretability, with only around 20% of samples rated as more preferable compared to the full model. Qualitative examples further reveal that this setting fails to produce coherent or meaningful explanations. Introducing the $RC$ loss alone improves interpretability by emphasizing a small set of critical concepts, resulting in more focused, though sometimes noisier, explanations. Conversely, the $C^3$ loss enhances the encoder's concept recognition ability, which slightly improves performance but fails to prioritize key concepts, leading to diffuse or less relevant explanations. When both $C^3$ and $RC$ losses are combined, the model achieves the best interpretability. The $C^3$ loss enhances concept recognition, while the $RC$ loss ensures the similarity is dominated by salient, human-aligned features. Although this configuration may slightly compromise performance metrics, it yields the most coherent and informative explanations across modalities.

Table 8: The zero-shot and retrieval comparison for ablation study of our proposed objectives.

| $C^3$ loss | RC loss | CUB | | MS-COCO | | | |
|:---:|:---:|:---:|:---:|:---:|:---:|:---:|:---:|
| | | R@1 | R@5 | T2I R@1 | T2I R@5 | I2T R@1 | I2T R@5 |
| | | 56.85% | 87.23% | 37.60% | 63.40% | 51.24% | 77.12% |
| ✓ | | 58.65% | 87.47% | 37.82% | 63.90% | 52.08% | 78.02% |
| | ✓ | 50.07% | 81.91% | 36.17% | 62.59% | 50.32% | 75.30% |
| ✓ | ✓ | 53.66% | 85.45% | 36.49% | 62.79% | 51.38% | 76.48% |

## 5 CONCLUSION

In this paper, we introduce CLIP-DMD, a task-agnostic and interpretable CLIP similarity framework based on Dual Modality Decomposition. By leveraging a Sparse Autoencoder, our method identifies and disentangles meaningful concepts embedded in CLIP features. The resulting concept representations are optimized using the InfoNCE loss to capture relationships across diverse image-text pairs. To further enhance interpretability, we propose two novel objectives: the Corpus Cycle Consistency loss, which reinforces concept recognition, and the Rate Constraint loss, which emphasizes the most salient concepts contributing to similarity. Extensive experiments demonstrate that CLIP-DMD offers deeper insight into the underlying semantics driving CLIP similarity, while retaining robustness and delivering competitive performance across zero-shot classification, retrieval, and linear probing tasks.

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

# A APPENDIX

## A.1 LIMITATION

Despite the numerous advantages, CLIP-DMD has a few potential limitations:

**Corpus limitation.** To interpret the learned corpus features in the decoder, we assign concept names based on a large predefined corpus consisting of unigrams and bigrams. However, this strategy limits semantic granularity, often resulting in only partially accurate or incomplete interpretations of the learned features. SpLiCE (Bhalla et al., 2024) faces a similar constraint, as its reliance on the same type of corpus restricts its ability to capture more nuanced or compositional concepts, as does DN-CBM (Rao et al., 2024).

To address this limitation, we explored using fixed predefined corpus embeddings as decoder weights to enforce direct interpretability. However, this approach significantly reduces the model's expressiveness and leads to noticeable degradation in both interpretability and task performance. To better balance these objectives, we adopt learnable concept weights in the decoder, allowing the model to discover more expressive and contextually relevant features while preserving competitive accuracy.

For future work, we plan to explore more advanced concept-to-feature mapping techniques that further enhance semantic precision without sacrificing performance, as well as investigate the use of LLMs to generate more diverse and expressive descriptions that better bridge the gap between model features and human understanding.

**Noise concepts.** While CLIP-DMD delivers strong performance and enhanced interpretability, the generated explanations may occasionally include noisy concepts, irrelevant terms that do not meaningfully contribute to identifying shared semantics across modalities. Further refinement in the decomposition process could help suppress such noise and yield more focused, high-fidelity explanations.

## A.2 CONCPET CORPUS GENERATION

We adopt the concept corpus introduced in SpLiCE (Bhalla et al., 2024), which is constructed by selecting the most frequent one-word and two-word bigrams from the text captions of the LAION-400M dataset. During preprocessing, NSFW samples are removed, and the remaining concepts are further pruned to ensure that no two concepts exceed a similarity threshold of 0.9, preventing redundancy within the vocabulary. The resulting corpus contains the 10000 most common single-word concepts and 5000 of the most common two-word concepts. This large and diverse corpus provides broad semantic coverage and serves as a strong foundation for evaluating concept-based decomposition methods like CLIP-DMD and SpLiCE.

## A.3 MORE QUANTITATIVE RESULTS

**Datasets.** Five additional datasets are adopted for our evaluation:
○ **Caltech101** (Li et al., 2022)contains 9145 images across 101 object categories, along with an additional background class. The categories span a wide range of objects, including animals, vehicles, instruments, and everyday items, with each class containing approximately 40 to 800 images (most having around 50). The images exhibit moderate scale variation, mild pose changes, and generally clean backgrounds. For our experiments, we randomly sample one-quarter of the images from each class to form the test set, using the remaining images for training.
○ **CIFAR100** (Krizhevsky et al., 2009) consists of 60000 images of size $32 \times 32$, distributed across 100 object categories. Each class contains 600 images, with 500 used for training and 100 for testing. The dataset encompasses a diverse range of everyday objects, categorized into 20 superclasses, including animals, vehicles, household items, and natural scenes. In our experiments, we use the fine-grained labels corresponding to all 100 categories.
○ **DTD** (Cimpoi et al., 2014) consists of 5640 images categorized into 47 describable texture attributes, such as striped, bumpy, veined, and zigzagged. Each category includes 120 images collected from in-the-wild environments, capturing substantial variation in scale, illumination, surface geometry, and imaging conditions. For all experiments, we follow the first official split provided with the dataset to construct the training and test sets.
○ **Flowers** (Nilsback & Zisserman, 2008) is a fine-grained visual classification dataset that contains 8189 images spanning 102 categories of flowers commonly found in the United Kingdom. Each class includes between 40 and 258 images, with notable variation in scale, pose, illumination, and background complexity. The dataset provides a standard split of 1020 training images, 1020 validation images, and 6149 test images. In our experiments, we utilize the training and validation sets due to the lack of a label for the testing set.
○ **PET** (Parkhi et al., 2012) is a fine-grained recognition benchmark containing 7349 images across 37 pet breeds, including 12 cat breeds and 25 dog breeds. Each class comprises roughly 200 images with substantial variation in pose, scale, occlusion, and lighting conditions. For all experiments, we combine the original training and validation sets into a single training set and evaluate performance on the test set.

As shown in Table 9, we additionally report zero-shot results on several datasets (Caltech101, CIFAR100, DTD, Flowers, PET), comparing CLIP-DMD against the CLIP and SpLiCE. These experiments demonstrate that the sparse features learned by CLIP-DMD, while reflecting concept-level structure, remain applicable for downstream tasks. The results show that CLIP-DMD consistently outperforms SpLiCE, another interpretable method whose performance drops substantially under the

Table 9: More zero-shot performance comparison on various models and datasets. The result shows that CLIP-DMD provides competitive discriminative concept-level sparse features compared to the other interpretable method, SpLiCE. For CLIP-DMD, we execute 3 times and report the mean and standard deviation.

| Model | Method | interpret-ability | Caltech101 | CIFAR100 | DTD | Flowers | PET |
|---|---|---|---|---|---|---|---|
| ResNet50 | CLIP | | 79.86 | 40.62 | 38.56 | 60.51 | 83.21 |
| | SpLiCE | ✓ | 0.33 | 0.08 | 0.05 | 0.00 | 0.05 |
| | CLIP-DMD | ✓ | $82.57_{\pm 0.68}$ | $39.75_{\pm 0.25}$ | $36.90_{\pm 0.66}$ | $49.88_{\pm 1.18}$ | $75.30_{\pm 0.29}$ |
| ViT-B-32 | CLIP | | 88.99 | 75.91 | 51.97 | 73.23 | 87.74 |
| | SpLiCE | ✓ | 0.47 | 0.39 | 0.27 | 0.49 | 0.30 |
| | CLIP-DMD | ✓ | $90.12_{\pm 0.48}$ | $72.87_{\pm 0.30}$ | $49.13_{\pm 0.98}$ | $64.75_{\pm 1.21}$ | $83.89_{\pm 0.78}$ |
| ViT-B-16 | CLIP | | 89.22 | 77.23 | 52.98 | 71.15 | 89.45 |
| | SpLiCE | ✓ | 0.52 | 0.08 | 0.05 | 0.00 | 0.19 |
| | CLIP-DMD | ✓ | $88.16_{\pm 0.73}$ | $75.05_{\pm 0.27}$ | $48.67_{\pm 1.87}$ | $66.18_{\pm 0.90}$ | $84.15_{\pm 0.10}$ |

Table 10: More linear probing performance comparison on various models and datasets. For each experiment, we execute three times and report the mean and standard deviation. The learned sparse features reflecting the composition of the concepts has the similar discriminability with CLIP resulting in comparable performance. (The best and the 2nd-best performances are denoted in red and blue color.)

| Backbone | Method | Caltech101 | CIFAR100 | DTD | Flowers102 | PET |
|---|---|---|---|---|---|---|
| ResNet50 | CLIP | $94.56\%_{\pm 0.03\%}$ | $69.89\%_{\pm 0.06\%}$ | $72.91\%_{\pm 0.17\%}$ | $97.80\%_{\pm 0.00\%}$ | $87.53\%_{\pm 0.04\%}$ |
| | SpLiCE | $37.00\%_{\pm 0.10\%}$ | $42.68\%_{\pm 0.06\%}$ | $57.96\%_{\pm 0.03\%}$ | $23.15\%_{\pm 1.19\%}$ | $52.04\%_{\pm 0.06\%}$ |
| | DN-CBM | $93.54\%_{\pm 0.10\%}$ | $68.72\%_{\pm 0.06\%}$ | $70.41\%_{\pm 0.08\%}$ | $95.64\%_{\pm 0.07\%}$ | $77.59\%_{\pm 0.03\%}$ |
| | CLIP-DMD | $93.41\%_{\pm 0.05\%}$ | $69.87\%_{\pm 0.03\%}$ | $71.99\%_{\pm 0.12\%}$ | $97.31\%_{\pm 0.21\%}$ | $86.37\%_{\pm 0.00\%}$ |
| ViT-B-32 | CLIP | $97.60\%_{\pm 0.17\%}$ | $85.75\%_{\pm 0.06\%}$ | $79.26\%_{\pm 0.21\%}$ | $98.66\%_{\pm 0.00\%}$ | $90.63\%_{\pm 0.07\%}$ |
| | SpLiCE | $39.37\%_{\pm 0.18\%}$ | $67.50\%_{\pm 0.05\%}$ | $63.23\%_{\pm 0.03\%}$ | $33.62\%_{\pm 2.12\%}$ | $59.78\%_{\pm 0.06\%}$ |
| | DN-CBM | $97.52\%_{\pm 0.03\%}$ | $86.00\%_{\pm 0.06\%}$ | $79.22\%_{\pm 0.11\%}$ | $98.17\%_{\pm 0.00\%}$ | $88.87\%_{\pm 0.04\%}$ |
| | CLIP-DMD | $97.58\%_{\pm 0.05\%}$ | $86.36\%_{\pm 0.02\%}$ | $79.73\%_{\pm 0.05\%}$ | $98.21\%_{\pm 0.07\%}$ | $90.21\%_{\pm 0.08\%}$ |
| ViT-B-16 | CLIP | $97.77\%_{\pm 0.12\%}$ | $86.13\%_{\pm 0.13\%}$ | $81.65\%_{\pm 0.09\%}$ | $99.10\%_{\pm 0.07\%}$ | $92.26\%_{\pm 0.14\%}$ |
| | SpLiCE | $39.50\%_{\pm 0.23\%}$ | $71.03\%_{\pm 0.04\%}$ | $64.66\%_{\pm 0.06\%}$ | $33.99\%_{\pm 1.48\%}$ | $65.40\%_{\pm 0.14\%}$ |
| | DN-CBM | $97.74\%_{\pm 0.12\%}$ | $86.53\%_{\pm 0.09\%}$ | $81.19\%_{\pm 0.08\%}$ | $98.82\%_{\pm 0.07\%}$ | $90.87\%_{\pm 0.03\%}$ |
| | CLIP-DMD | $97.69\%_{\pm 0.05\%}$ | $86.58\%_{\pm 0.07\%}$ | $81.56\%_{\pm 0.11\%}$ | $98.90\%_{\pm 0.00\%}$ | $92.82\%_{\pm 0.02\%}$ |

zero-shot setting. Although CLIP-DMD exhibits some performance degradation compared to CLIP, it uniquely offers concept-level explanations through its sparse features, an interpretability capability that the original CLIP model does not provide.

We also provide the additional result on linear probing to evaluate the discriminability of sparse features extracted by CLIP-DMD compared to other methods. The linear probing model consists of a single linear layer and is trained for 100 epochs, optimized with AdamW and set learning rate $10^{-2}$, weight decay $10^{-4}$ for every dataset. As shown in Table 10, CLIP-DMD achieves performance comparable to CLIP across various models and datasets. In contrast, DN-CBM exhibits performance drops in some settings (DTD, Flowers102 on ResNet50, and all models on PET), and SpLiCE fails to match the performance of either the CLIP or CLIP-DMD.

## A.4 MORE QUALITATIVE RESULTS

In this section, we present additional quantitative results to assess the robustness and interpretability of CLIP-DMD across different backbone architectures (e.g., ViT-B/16 and ResNet50). As illustrated in Figures 7 and 8, CLIP-DMD consistently delivers meaningful insights into CLIP similarity, regardless of the underlying backbone, demonstrating the portability and generalizability of our approach. Additionally, Figure 9 presents a broader range of examples compared to SpLiCE (Bhalla et al., 2024), highlighting CLIP-DMD's superior ability to explain image-caption similarity through concept-level interpretations. Moreover, Figures 10, 11, and 12 display the top-responding images for each discovered concept across architectures. These results further emphasize the robustness and

consistency of concept alignment in CLIP-DMD, reinforcing its effectiveness across diverse model backbones.

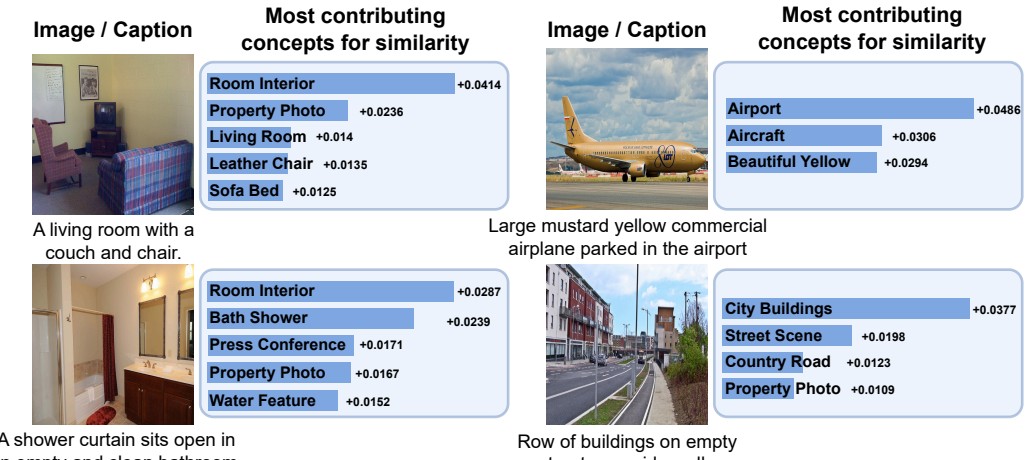

Figure 7: The explanations generated by CLIP-DMD to illustrate why a given image and caption are considered similar. (Backbone: ResNet50)

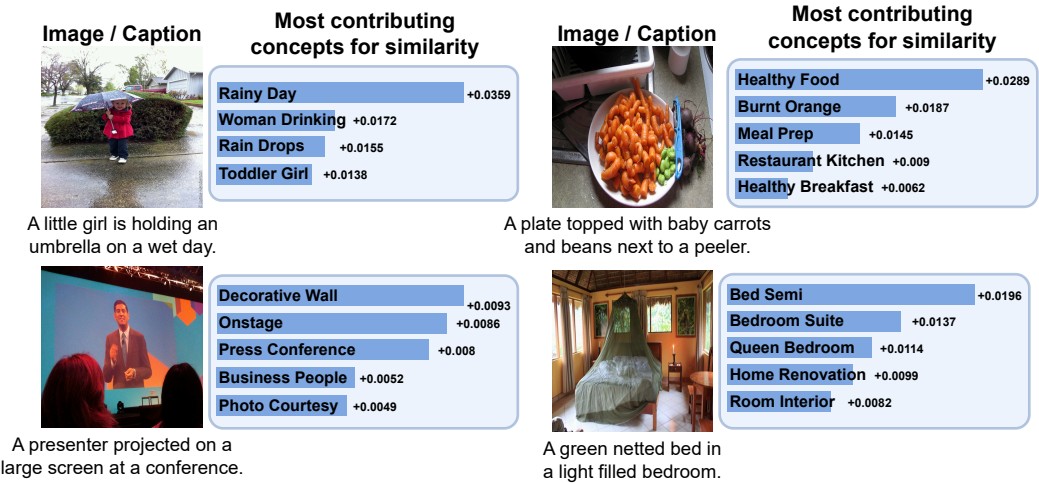

Figure 8: The explanations generated by CLIP-DMD to illustrate why a given image and caption are considered similar. (Backbone: ViT-B-16)

## A.5 QUANTITATIVE EVALUATION

**How we evaluate on zero-shot and retrieval performance?** To demonstrate that the learned sparse concept features not only capture semantic factors but also support downstream tasks, we evaluate their quantitative performance. Specifically, we compute similarity scores based on these features for zero-shot classification (image-to-label) and retrieval (image-to-caption), using the following formulation:

$$\mathbf{S}(\mathcal{F}_{img}^{CLIP}, \mathcal{F}_{text}^{CLIP}) = \mathbf{Cos}(\mathcal{F}_{img}^{CLIP}, \mathcal{F}_{text}^{CLIP}). \tag{7}$$

We redefine similarity using the sparse concept features, which represent the underlying concept composition, as follows:

$$\mathbf{S}(\mathcal{F}_{img}^{s}, \mathcal{F}_{text}^{s}) = \mathbf{Cos}(\mathcal{F}_{img}^{s}, \mathcal{F}_{text}^{s}). \tag{8}$$

This approach measures similarity at the concept level, rather than relying on high-dimensional, non-interpretable features. As a result, high similarity is achieved only when two instances share similar

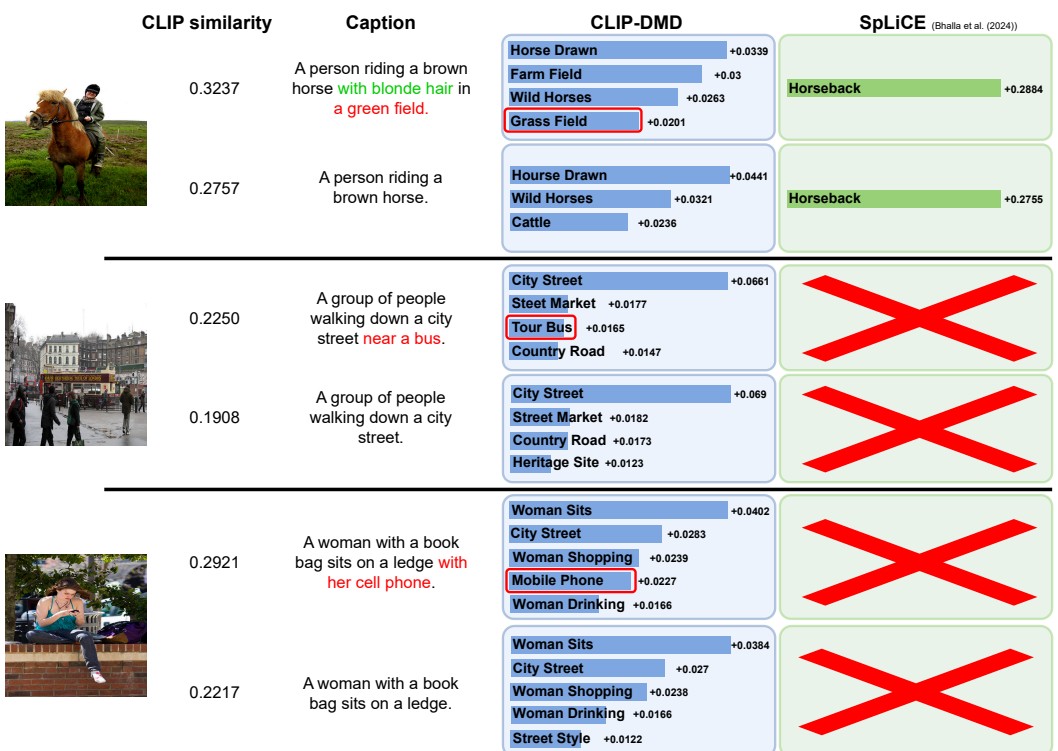

Figure 9: The examples demonstrate that CLIP-DMD effectively explains disparities in CLIP similarity, whereas SpLiCE (Bhalla et al., 2024) fails to provide meaningful justification. In each case, CLIP-DMD accurately captures critical semantic differences and clearly reflects them in its generated explanations, e.g highlighting "grass field" in the first example, "tour bus" in the second, and "mobile phone" in the third. In contrast, SpLiCE does not capture or reflect these key elements, leading to less informative or incomplete explanations. For the second and third cases, SpLiCE can't even point out the concept to form the explanation. (Backbone: ViT-B-32)

conceptual compositions. To ensure a fair comparison across methods, we evaluate performance using the same feature representations that are used to generate explanations. Specifically, for ours and SpLiCE (Bhalla et al., 2024), we employ the corresponding sparse concept features to assess zero-shot classification, retrieval, and linear probing performance.

**Why we only compare to SpLiCE?** We limit our comparison to SpLiCE (Bhalla et al., 2024) for zero-shot and retrieval performance because other methods either focus solely on single-modality interpretation or are designed for basic image classification tasks, making them lack the capability to measure the similarity between image and text. Although DN-CBM (Rao et al., 2024) supports both image and text modalities, its sparse features are not projected into a shared concept space. As a result, feature dimensions from the two modalities do not carry consistent semantic meanings and cannot be directly compared to compute cross-modal similarity. In contrast, SpLiCE uses a unified decomposition basis derived from a shared corpus, ensuring that the same feature dimensions represent the same concepts across both modalities, making it suitable for cross-modal similarity evaluation.

## A.6 ADDITIONAL ABLATION STUDY

In this section, we present additional ablation studies. We first examine the effect of the parameter $M$, which controls the dimensionality of the sparse concept features. As shown in Table 11, CLIP-DMD consistently achieves strong and superior performance compared to SpLiCE (Bhalla et al., 2024) across different values of $M$. Furthermore, Table 12 demonstrates that CLIP-DMD maintains higher and more stable interpretability across varying dimensions than SpLiCE. We then analyze the effect

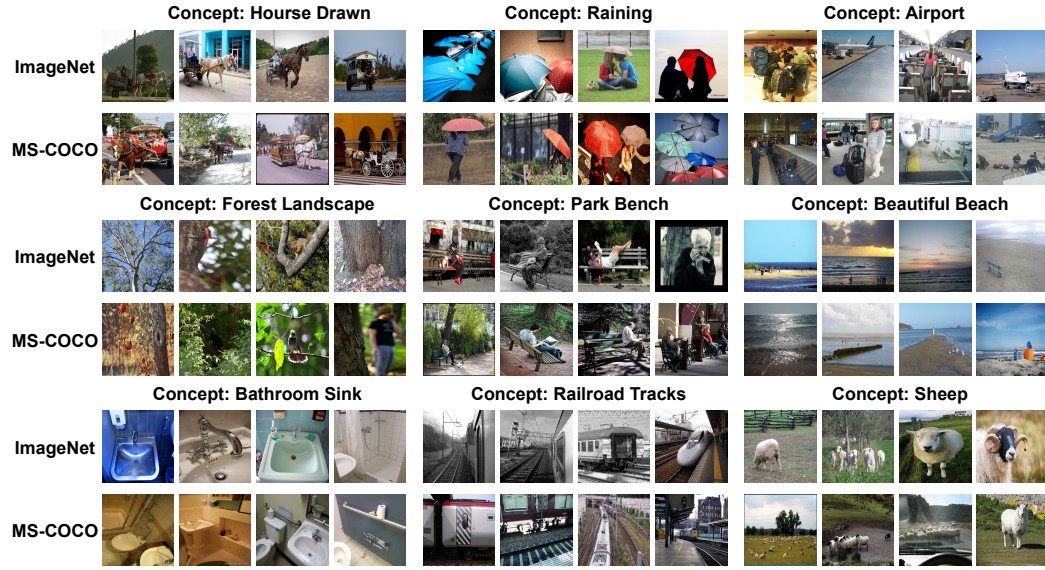

Figure 10: The sampled concepts demonstrate that the features learned by CLIP-DMD consistently and accurately capture the corresponding semantic concepts. (Backbone: ResNet50)

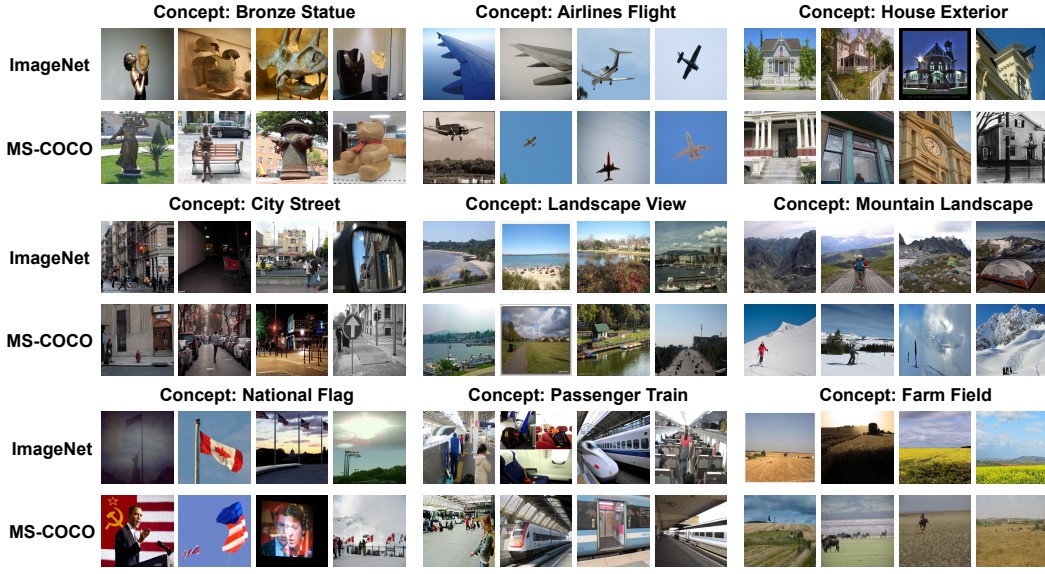

Figure 11: The sampled concepts demonstrate that the features learned by CLIP-DMD consistently and accurately capture the corresponding semantic concepts. (Backbone: ViT-B-32)

of $k$ in the $RC$ loss, as reported in Tables 13 and 14. The results indicate that $k = 32$ provides strong performance in zero-shot, retrieval, and interpretability. Nonetheless, $k$ can be tuned based on task requirements, for instance, setting $k = 256$ yields higher accuracy on fine-grained tasks (e.g., CUB).

We also analyze the role of the InfoNCE loss. As shown in Table 15, removing the InfoNCE loss causes CLIP-DMD to fail, since the resulting sparse features lack sufficient discriminability across cases. This underscores the critical importance of InfoNCE in our method.

Finally, we provide evidence supporting the use of two separate encoders for the image and text modalities rather than a single shared encoder. In practice, employing distinct decoders to learn sparse features for each modality leads to superior performance. This finding motivates our adoption of the separate-decoder design, as demonstrated in Table 16.

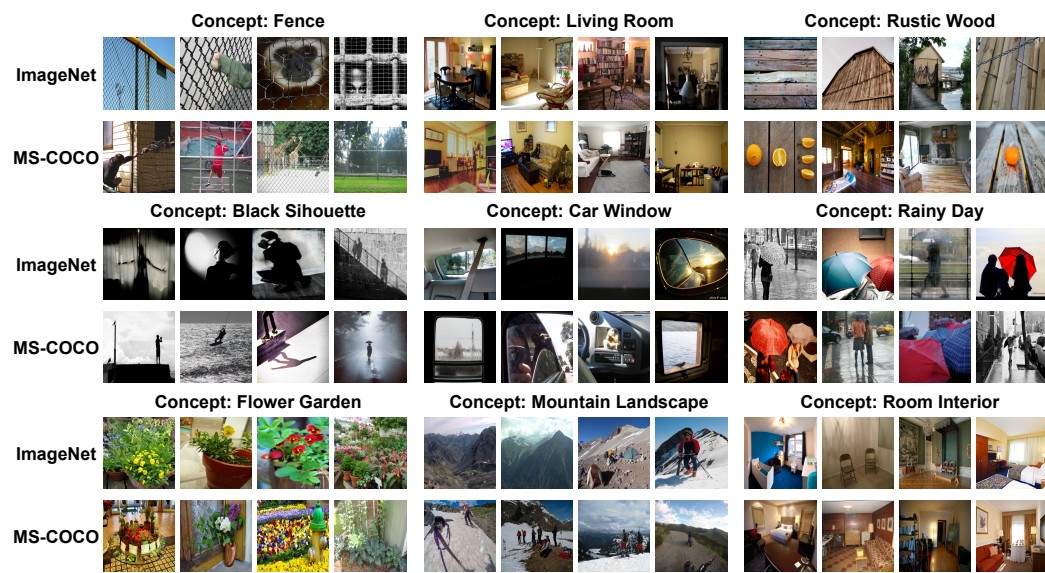

Figure 12: The sampled concepts demonstrate that the features learned by CLIP-DMD consistently and accurately capture the corresponding semantic concepts. (Backbone: ViT-B-16)

Table 11: The zero-shot and retrieval comparison for ablation study of $M$ (the dimensionality of sparse features). (Backbone: ViT-B-32)

|  | | CUB | | ImageNet | | MS-COCO | | | |
|  | | R@1 | R@5 | R@1 | R@5 | T2I R@1 | T2I R@5 | I2T R@1 | I2T R@5 |
|---|---|---|---|---|---|---|---|---|---|
| | 4096 | 53.21% | 85.04% | 61.32% | 87.21% | 36.40% | 62.68% | 50.84% | 75.70% |
| The $M$ in | 8192 | 53.43% | 85.67% | 61.71% | 87.45% | 36.45% | 62.85% | 51.12% | 76.26% |
| CLIP-DMD | 10240 | 53.68% | 84.88% | 61.89% | 87.64% | 36.67% | 62.84% | 51.56% | 76.32% |
| | 15360 | 53.66% | 85.45% | 61.80% | 87.53% | 36.49% | 62.79% | 51.38% | 76.48% |
| SpLiCE (Bhalla et al., 2024) | | 7.34% | 26.48% | 34.95% | 60.90% | 6.04% | 16.59% | 10.60% | 25.46% |

Table 12: The interpretation ablation study of the sparse feature dimension. We compare the different dimensions of our method to SpLiCE (Bhalla et al., 2024) as the baseline (fixed dimension) to compare the capability of interpretation with different dimensions. (Backbone: ViT-B-32)

| | Interpretability | | | |
| Dimension | 4096 | 8192 | 10240 | 15360 |
|---|---|---|---|---|
| CLIP-DMD (Ours) | 57.93% | 59.59% | 58.38% | 60.09% |
| SpLiCE (Bhalla et al., 2024) | 42.07% | 40.41% | 41.62% | 39.91% |

Table 13: The zero-shot and retrieval comparison for ablation study of $k$ (for the $RC$ loss). (Backbone: ViT-B-32)

| $k$ | CUB | | ImageNet | | MS-COCO | | | |
| | R@1 | R@5 | R@1 | R@5 | T2I R@1 | T2I R@5 | I2T R@1 | I2T R@5 |
|---|---|---|---|---|---|---|---|---|
| 32 | 53.66% | 85.45% | 61.80% | 87.53% | 36.49% | 62.79% | 51.38% | 76.48% |
| 64 | 53.24% | 85.26% | 61.21% | 87.33% | 36.70% | 62.86% | 51.42% | 76.38% |
| 128 | 51.71% | 85.02% | 61.04% | 87.08% | 36.32% | 62.37% | 50.96% | 76.14% |
| 256 | 54.00% | 85.24% | 61.68% | 87.35% | 36.30% | 62.52% | 51.14% | 76.46% |

Table 14: The interpretation ablation study of $k$ (for the $RC$ loss). (Backbone: ViT-B-32)

| $k$ | Interpretability | | | |
|---|---|---|---|---|
| | 32 | 64 | 128 | 256 |
| CLIP-DMD (Ours) | 60.09% | 59.77% | 59.91% | 59.93% |
| SpLiCE (Bhalla et al., 2024) | 39.91% | 40.23% | 40.09% | 40.07% |

Table 15: The ablation study on the InfoNCE loss shows that, without it, the sparse features lose discriminability across cases, leading to poor performance in zero-shot and retrieval tasks.(Backbone: ViT-B-32)

| InfoNCE | CUB | | ImageNet | | MS-COCO | | | |
|---|---|---|---|---|---|---|---|---|
| | R@1 | R@5 | R@1 | R@5 | T2I R@1 | T2I R@5 | I2T R@1 | I2T R@5 |
| | 1.02% | 6.01% | 2.61% | 9.51% | 2.96% | 9.89% | 1.10% | 4.12% |
| ✓ | 54.03% | 85.49% | 61.98% | 87.51% | 36.64% | 62.81% | 51.25% | 76.35% |

Table 16: The ablation study on the encoder shows that using distinct encoders for the image and text modalities yields a slight performance improvement.(Backbone: ViT-B-16)

| | CUB | | ImageNet | | MS-COCO | | | |
|---|---|---|---|---|---|---|---|---|
| | R@1 | R@5 | R@1 | R@5 | T2I R@1 | T2I R@5 | I2T R@1 | I2T R@5 |
| 1 encoder | 57.29% | 88.91% | 65.65% | 89.76% | 39.24% | 65.08% | 53.28% | 76.25% |
| 2 encoders | 57.79% | 88.29% | 64.80% | 89.22% | 39.43% | 65.44% | 54.05% | 78.08% |

## A.7 ADDITIONAL COMPARISON

We additionally report linear probing results against Matryoshka SAE (MSAE) (Zaigrajew et al., 2025), a recent baseline that learns hierarchical representations in SAE to jointly enhance reconstruction quality and sparsity; however, it still processes image and text modalities separately. As shown in Table 17, our CLIP-DMD outperforms MSAE. The discrepancy between the reported performance and that in the original paper arises from differences in the accuracy calculation. Specifically, MSAE evaluates alignment between the reconstructed and original CLIP features, reflecting the amount of information preserved in reconstruction. In contrast, our evaluation measures how effectively the sparse features, capturing the underlying concepts of the image or text, can distinguish between cases. The reason that we only report linear probing results is that it decomposes image and text features separately, leading to a non-unified concept space, which is the same reason as DN-CBM (Rao et al., 2024).

Table 17: The linear probing comparison with MSAE(Zaigrajew et al., 2025).

| Backbone | Method | CUB | TinyImageNet |
|---|---|---|---|
| ViT-B-16 | MSAE(Zaigrajew et al., 2025) | 0.57 ±0.03 | 0.61 ±0.02 |
| | CLIP-DMD (Ours) | 78.12 ±0.05 | 77.47 ±0.04 |
| ViT-L-14 | MSAE(Zaigrajew et al., 2025) | 0.45 ±0.02 | 0.59 ±0.03 |
| | CLIP-DMD (Ours) | 84.04 ±0.04 | 83.13 ±0.02 |

## A.8 LLM EVALUATION

### A.8.1 SINGLE MODEL

In this section, we describe how we use a Large Language Model (LLM) to evaluate interpretability. We first extract the sparse features that represent the concept composition of the image and text CLIP features, and compute the cross-modal similarity using these concept-level sparse features. This information forms a group of explanations consisting of two components:(1) the set of shared

Table 18: The interpretability comparison between CLIP-DMD and SpLiCE on the MS-COCO dataset, along with the human-alignment rate of each evaluated model, is summarized in the table.

| Evaluated Model | Interpretation Comparison | | Human Alignment Rate |
| --- | --- | --- | --- |
| | CLIP-DMD (Ours) | SpLiCE | |
| LLama-3.2-11B-Vision-Instruct | 62.75 $\pm$ 2.43 | 37.25 $\pm$ 2.43 | 59.00 $\pm$ 8.00 |
| Gemma-3-27B-it | 71.07 $\pm$ 0.07 | 28.93 $\pm$ 0.07 | 67.00 $\pm$ 2.00 |
| Qwen2.5-VL-32B-Instruct | 53.70 $\pm$ 0.32 | 46.30 $\pm$ 0.32 | 59.00 $\pm$ 3.00 |
| LLama-3.2-11B-Vision-Instruct + Gemma-3-27B-it + Qwen2.5-VL-32B-Instruct | 64.16 $\pm$ 0.13 | 35.84 $\pm$ 0.13 | 72.00 $\pm$ 2.00 |

*Each experiment is conducted three times, and the results are reported as the mean and standard deviation to account for randomness introduced by permutation and ordering.

concepts between the image and the caption, generated by the interpretability method, where we retain only the most influential concepts, those that accumulated account for at least 30% of the sparse similarity by picking in sorted order from largest to smallest; and (2) the similarity score computed using the sparse concept features. Next, we present two sets of explanations (one from each interpretability method), along with the source image and caption, to the LLM and ask it to evaluate their quality. The overall prompting setup is illustrated in Figure 13. To avoid positional bias (e.g., favoring the first-listed option), we instruct the LLM to assign a score to each explanation rather than directly selecting a preferred one. Additionally, we randomly permute the order of the options to further mitigate ordering bias. Each experiment is repeated three times to account for randomness in scoring behavior. After processing all samples in the dataset, we compute the proportion of cases in which each method receives the highest score each time. This score rate serves as the final interpretability evaluation metric which we report the mean and standard deviation.

Figures 14 and 15 present examples of LLM responses along with their explanations. For the actual evaluation, we additionally instructed the LLM to output only the scores in the format [*Score1* : *Score2*] to reduce the computation. After evaluating all samples, we counted how often each method received the highest score and reported these counts as ratios in Table 18. The results show that various LLM models prefer CLIP-DMD to SpLiCE. Especially, the Gemma-3-27-it has the highest agreement among the three models and achieves the highest human alignment rate, which further enhances the reliability of the evaluation result.

### A.8.2 ENSEMBLE MODELS

To further strengthen the reliability of the evaluation and reduce potential bias from any single model, we introduce an ensemble LLM evaluation, where multiple LLMs collectively serve as expert judges. By aggregating the decisions from several LLMs, the final outcome becomes less sensitive to the idiosyncrasies or biases of any one model, resulting in a more balanced and diverse assessment of interpretability. Following the same scoring procedure used for the single-LLM setting, each LLM independently assigns scores to the explanations. We then aggregate their scores and determine the winning method for each sample through majority voting. Finally, we report the percentage of cases in which each method receives the most votes. To ensure robustness, we repeat the entire evaluation three times for each LLM and report the mean and standard deviation across runs. As shown in Table 18, the ensemble model achieves the highest human alignment rate compared to the single model and presents a preference for CLIP-DMD.

### A.8.3 HUMAN ALIGNMENT

To assess whether the LLM's judgments align with human reasoning, we measure the agreement rate between LLM decisions and human evaluations. We randomly sample 50 cases from the dataset as the evaluation set. Using the same information provided to the LLM, we ask human participants, individuals who are currently pursuing or have completed a graduate degree in a STEM field and are familiar with machine learning concepts, to directly choose the interpretation they find most reasonable for explaining the CLIP similarity based on the given concepts and similarity scores. We

Given a picture, a sentence describing the image, and CLIP similarity between the image and the sentence reflecting how well the two modalities match, score each group of CLIP similarity explanations.
The scoring process is based on considering whether the given explanation can show the relation strength given by CLIP similarity across the image and the sentence, and whether the similarity from the explanation is close to CLIP similarity, which gets a high score if the similarity from the explanation is close to CLIP similarity.
Each method provides explanations by listing the shared concepts between the image and the sentence, and the similarity calculated by the generated explanations.
The explanation might also contain unrelated concepts to the image and the sentence that share the same idea, which should consider the higher-level meaning of these concept.
Typically, the CLIP similarity between the image and the sentence of around 0.3 will be treated as an acceptable match, similarity between 0.1 and 0.2 will be treated as a partial match, which might match with partially related concepts, such as a woman in the image might list women shopping, and similarity between 0 and 0.1 will be treated as non-related pair which the explanation will list more irrelevant noise concepts.
If the explanation lists more partially related concepts, the score should be higher than the one listing fewer partially related concepts.
If the explanation captures the broader context and relationships between the concepts in images and sentences, the score should be higher.
The scoring process should not only consider the foreground information but also the background information.
Avoid giving two groups the same scores.
The score is in the range of 1 to 10, with only integers. 1 refers to the explanation that can't reflect the CLIP score or doesn't show any concept, e.g., None. 10 refers to the explanation that can reflect the CLIP score.
Caption: {}
Similarity : {:.6f}
===
The first group explains:
Concepts: {}
Similarity: {}

The second group explains:
Concepts: {}
Similarity: {}
===
# Only report two gropus scores in the format in the format [x:y] where x, y refer to first group score and second group score respectively.

Figure 13: The LLM prompt template for evaluating the interpretation.

then compare these human choices with the decisions made by the LLM (evaluated three times to account for randomness), compute the alignment rate, and report the mean and standard deviation.

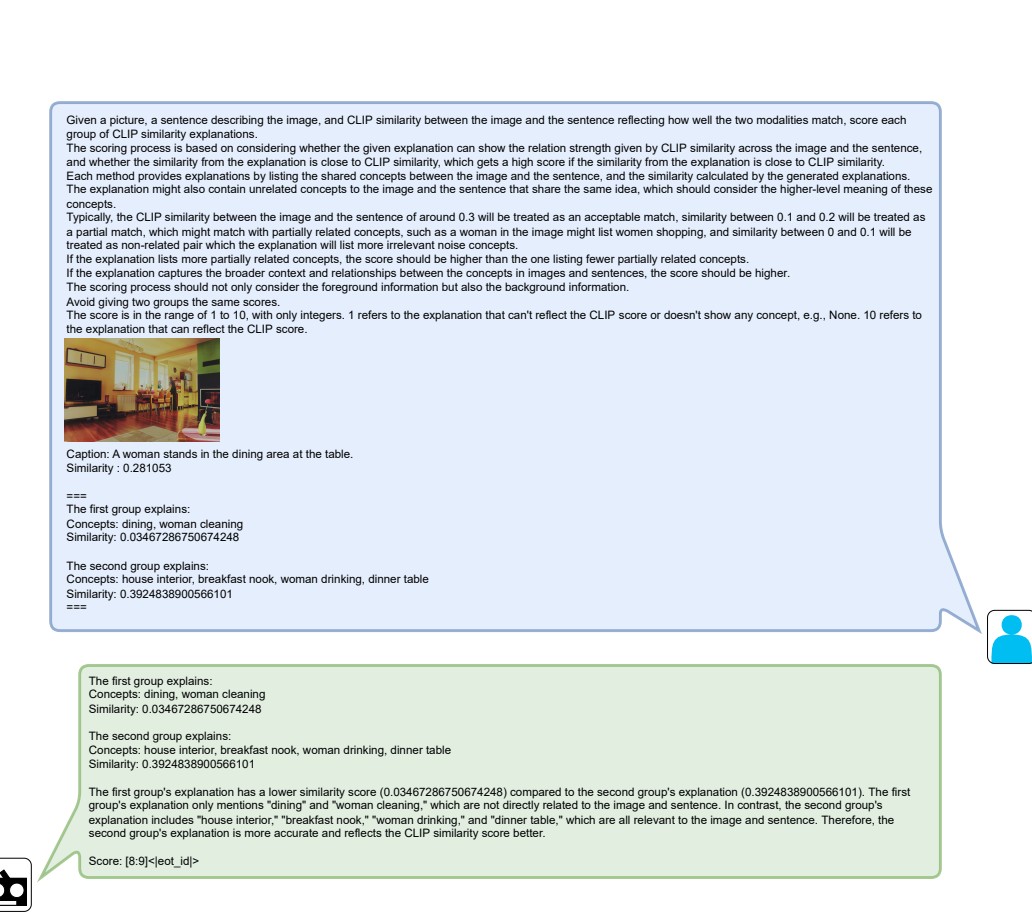

Figure 14: An example of the LLM-based evaluation is shown here. By inserting the generated concepts and the similarity scores (computed from sparse features) into a prompt template, the LLM evaluates and assigns a score to each explanation group. In this example, the first and second groups correspond to SpLiCE (Bhalla et al., 2024) and CLIP-DMD, respectively. According to the LLM's response, CLIP-DMD's explanation covers broader and more relevant concepts shared between the image and caption, while also yielding a similarity score closer to that of CLIP. In contrast, SpLiCE provides a narrower set of concepts and a much lower similarity score, making its explanation comparatively less informative.

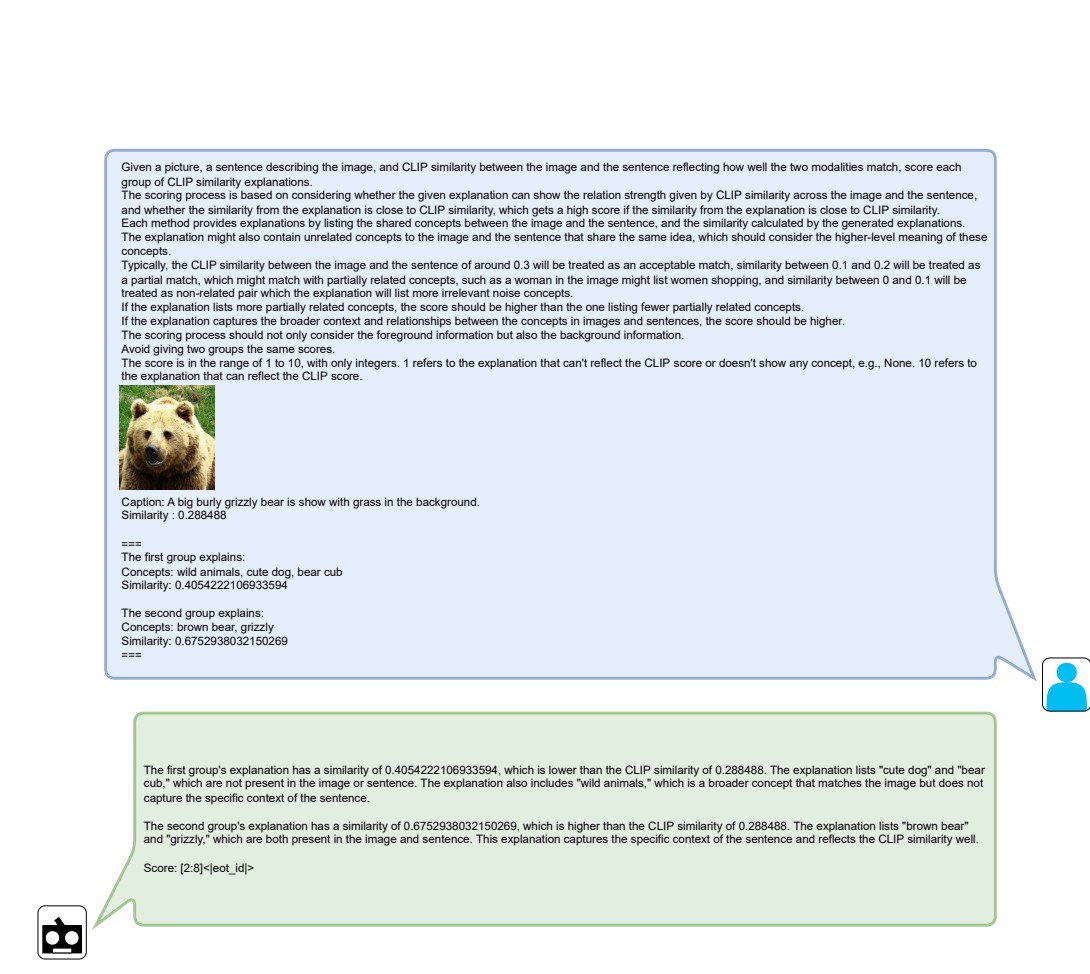

Figure 15: This is an example of the LLM-based evaluation. In this case, the first and second groups correspond to CLIP-DMD and SpLiCE (Bhalla et al., 2024), respectively. Although CLIP-DMD produces a similarity score closer to the original CLIP similarity, its explanation includes relatively noisy concepts, leading to a lower evaluation score. In contrast, SpLiCE provides more precise and relevant concepts, despite its similarity score being much higher than that of CLIP. This example demonstrates that the LLM balances both the quality of the concepts and the alignment of similarity when assigning scores.

## B  USING LLM FOR WRITING

We leverage a Large Language Model (LLM) to refine the writing, improving both fluency and clarity. After obtaining the refined sentences from the LLM, we further revise them to ensure closer alignment with the original technical content.

