# OpenReview forum: "Uncovering the Why: Interpretable CLIP Similarity via Dual Modalities Decomposition"
_ICLR.cc/2026/Conference — Submitted to ICLR 2026_

### Official Review · Reviewer_PbAs · 2025-10-25

**Soundness:** 2
**Presentation:** 3
**Contribution:** 3
**Rating:** 4
**Confidence:** 3

**Summary:**

The authors present a method to get at interpretability of CLIP features from a dual modality perspective, claiming that prior methods have been too focused on single modalities. The main comparison of methods is with another model called SpLiCE). Overall, the method described claims to more effectively use shared features to gain a deeper level of interpretability.

I will admit that my experience with CLIP feature interpretability research is not the best and It’s been over a year since I last really got into other work on the topic. However, I still feel like a good paper should clearly ground itself and prior research a bit better. There were moments when I felt my lack of understanding was to blame, but upon revisiting the difficult sections I’ve come to see that I just don’t think the paper explains it that well, particularly in explaining the related work and setting the context of the current proposal among other relevant literature. I also found the lack of extensive reporting of results to be quite deficient in backing up the claims. It's not clear to me there is evidence that stretches beyond, "Look at these images in the figure, and believe us that they are not cherry picked and they extend across all images."

**Strengths:**

The paper's strengths are the motivating factors in being more encompassing with regard to looking at joint structures when decomposing interpretable elements in CLIP. The modelling process and inspiration on the Concept Bottleneck Framework are good. There is a sound basis building on (limitedly-described) relevant literature and from what I can tell from the often-vague descriptions, there are sparks of interesting ideas, but while some areas of the paper are well-explained, this standard isn't maintained throughout.

**Weaknesses:**

Figure 1.  - What is this and what is it showing me?
Why are there no labels for what the values mean? I can see things are labelled as to going up and down but I’m not sure what the numbers actually mean. I’m more confused about the approach after having seen Figure 1 than before, which I think is opposite to the intended effect. The caption is not doing the job it is supposed to and the figure needs to be addressed. I do not see how “sea coast” and “beautiful beach” clearly contribute to explainability.
The paper repeatedly points to cherry-picked visual examples and claims that increased interpretability is “obvious” or heavily implied. I also see no sign of a single statistical test or hypothesis being statistically evaluated. For example: ”As shown in Figure 4, our method effectively identifies shared concepts across both modalities, providing a clear and interpretable rationale for the resulting CLIP similarity.”  - This in itself without any further (non-anecdotal) backing is not sufficient to back up the strong claim being made here. I don’t know how this generalises across more examples and I don’t know how cherry picked this example is.

I’m not claiming that the method is incorrect and that it could be truthful, but the burden of proof is on the authors to demonstrate this in a convincing manner and I am concerned that the authors think that these single examples are sufficient to convince readers that broadly, on the whole, across whole datasets, that these benefits must always hold. The evidence isn’t being presented to back up such claims.

The details of experiments that do seem more quantitative are very under-specified. I followed the trail across references from section 4 back to section 3.3 and then jumped to the appendix for more info, but the details on avoiding positional bias seem to be more suggested at rather than sufficiently documented. The section in the appendix on quantitative evaluation is a very vague set of statements that does not reach anywhere near the level of academic rigour required for acceptance at ICLR unfortunately. You can’t just put 3 images in a figure and make a few comments about them and call this a “quantitative analysis”.

Overall, I think the authors have an interesting idea and they could be onto something and I encourage them to continue this work. However, there are severe, severe shortcomings when it comes to presenting the results successfully in an academic submission. There is only a direct comparison to a single other method (SpLiCE) and many experimental methodological details are completely absent. To pick on one such example, when avoiding presentation order bias it is mentioned there are permutations of method orderings during evaluation. What does this mean if you’re not saying how many permutations were run and not showing averages or variability under these multiple runs? Or do the authors mean they make a permutation and treat the sample results the same. The results for these don’t seem to be anywhere (unless I’ve severely misunderstood something).

**Questions:**

1. Can you better characterise Figure 1 for me and create a better caption that explains the numbers and why some "increases" and some "decreases" seem to be all positive news for the proposed method? I just didn't get this part at ll.

2. Can you please revisit and add in some technical details on experiments that you performed that go beyond just looking at specific examples. The LLM evaluation just doesn't have enough detail for me to see that a successful attempt has been made at this.

3. I think you have a high chance at convincing me to increase my rating if you can completely rewrite your experimental section, have someone not familiar with your specific experiment look over it and see if they can fully follow the logic behind the experiments and they can provide feedback as to what is missing. From reading the paper, I fully get the sense that because the authors know all the details of the experiment, they are relying on implicit background knowledge that they haven't realised isn't present in the paper, and this causes confusion that the authors might not be able to appreciate.

4. Would you be able to provide an additional section in the appendix that can flesh out the way you derived the corpus of semantic concepts. This is absolutely a crucial central element to many of the experiments you ran and I need to understand how you create it a bit better because it's not explained in sufficient detail in your paper. A nice Figure would be most welcome, but a sufficient text explanation would also suffice.

---

> ### Author Response · Authors · 2025-11-26
>
> We thank the reviewer for taking the time to provide valuable feedback. In the spirit of continually improving the quality of our work, we address each comment in turn, starting with the identified weaknesses and then responding to the specific questions.
>
> ## Weakness 1:
> [*The meaning of figure 1*]
>
> The core idea of Figure 1 is to illustrate that the original CLIP model only outputs a similarity score between an image and a caption, without revealing how this similarity is derived. Although SpLiCE provides modality-specific concepts, the overlapping concepts and their computed similarity cannot explain why CLIP assigns a higher similarity to caption 2 than to caption 1, even yielding the opposite conclusion. In contrast, CLIP-DMD successfully figures out the core concept between the two pairs. By identifying the sea-related concept, our method explains why caption 2 achieves higher similarity to the image than caption 1. The similarity computed by considering these additional concepts aligns with CLIP’s behavior, correctly reflecting the higher similarity for caption 2.
>
> ## Weakness 2:
> [*The paper repeatedly points to cherry-picked visual examples and claims that increased interpretability is “obvious” or heavily implied.*]
>
> We also provide the LLM evaluation to verify the interpretability between CLIP-DMD and SpLiCE and the LLM decision alignment to human decisions.

---

> > ### Author Response · Authors · 2025-11-26
> >
> > ## Weakness 3:
> > [*Detail of quantitative experiments and the purpose*]
> >
> > - **For the zero-shot and retrieval task**:
> > To evaluate whether the learned sparse features can accurately capture cross-modal similarity, we test their performance on zero-shot classification and retrieval tasks. For both CLIP-DMD and SpLiCE, we extract sparse features for images and texts and apply them using the same evaluation protocol as the original CLIP model. Specifically, zero-shot classification is performed by pairing each class name with the prompt “a photo of a {class_name}” and assigning the image to the class whose anchor it is most similar to. For image-to-text (I2T) or text-to-image (T2I) retrieval tasks, we identify the most similar text (for I2T task)  or image (for T2I task) to the target query under the sparse feature space.
> >
> > - **For linear probing**:
> > The linear probing experiment evaluates the discriminative power of the sparse features by testing how well they can be separated using a simple linear classifier. We extract image sparse features from CLIP-DMD, SpLiCE, and DN-CBM, and train a model consisting of a single linear layer for 100 epochs using the AdamW optimizer with a learning rate of 1e-2 and weight decay of 1e-4. Higher linear probing accuracy indicates that the extracted sparse features are more distinguishable, making them easier for a simple linear model to classify.
> >
> > - **For Spearman’s rank correlation**: Spearman’s rank correlation evaluates how closely the similarity trends produced by the sparse features align with those of the original CLIP model. This metric assesses whether the sparse features can faithfully capture CLIP’s similarity behavior from a concept-level perspective. For this experiment, we use the MS-COCO dataset, where each image is paired with five captions. For each method (CLIP-DMD and SpLiCE), we compute the similarity between an image and its five captions, rank the captions according to similarity, and then compare this ranking to CLIP’s ranking using Spearman’s rank correlation. A higher correlation indicates that the similarity trends derived from the sparse features more closely follow those of CLIP.
> >
> > - **LLM evaluation**: To evaluate interpretability on a large dataset without extensive manual effort, we introduce a LLM-based evaluation procedure in which a vision–language model (LLM) judges the quality of interpretations generated by CLIP-DMD and SpLiCE. For each method, we construct an interpretation by gathering the sparse similarities computed from their sparse features, the overlapping concepts between the image and text, and retaining only the most influential concepts, those that collectively contribute at least 30% of the sparse similarity. These interpretations, along with the corresponding image, caption, and CLIP similarity score, are then provided to the LLM, which determines which one is more suitable. To avoid positional bias (e.g., favoring the first option), we ask the LLM to score each method independently rather than choose a preferred one. We also randomize the order of the method presentations to further reduce potential ordering effects. Interpretability is assessed on the MS-COCO dataset, which contains images paired with multiple captions. Using 5,000 samples, we collect the scores assigned by the LLM to both methods and compute the proportion of times each method obtains the highest score. This evaluation is repeated three times, and we report the mean rate as the final interpretability metric.
> >
> > - **For inference time**: To test the inference time, we measure the time each method takes to extract the input to the end of outputting the sparse feature. We average the time over 1000 executions and report the result.

---

> ### Author Response · Authors · 2025-11-26
>
> ## Weakness 4:
> [*The only comparison to the SpliCE*]
>
> We primarily compare our approach with SpLiCE because, to the best of our knowledge, other related methods (e.g., DN-CBM[1], PCBM[2], Res-CBM[3], OpenCBM[4], Grad-ECLIP[5]) do not provide single-modal explanations without cross-modal concept alignment, nor do they focus on interpreting similarity by identifying the most influential components contributing to the similarity score. Due to these fundamental differences in interpretability objectives and mechanisms, a direct comparison would not be well aligned.
>
> ## Question 1:
> The core idea of Figure 1 is to illustrate that the original CLIP model only outputs a similarity score between an image and a caption, without revealing how this similarity is derived. Although SpLiCE provides modality-specific concepts, the overlapping concepts and their computed similarity cannot explain why CLIP assigns a higher similarity to caption 2 than to caption 1, even yielding the opposite conclusion. In contrast, CLIP-DMD successfully figures out the core concept between the two pairs. By identifying the sea-related concept, our method explains why caption 2 achieves higher similarity to the image than caption 1. The similarity computed by considering these additional concepts aligns with CLIP’s behavior, correctly reflecting the higher similarity for caption 2.
>
> [1] Sukrut Rao, Sweta Mahajan, Moritz B¨ ohle, and Bernt Schiele. Discover-then-name: Task-agnostic
> concept bottlenecks via automated concept discovery. In Proceedings of the European Conference
> on Computer Vision (ECCV), 2024.
> [2] Mert Yuksekgonul, Maggie Wang, and James Zou. Post-hoc concept bottleneck models. In Interna-
> tional Conference on Learning Representations (ICLR), 2023.
> [3] Chenming Shang, Shiji Zhou, Hengyuan Zhang, Xinzhe Ni, Yujiu Yang, and Yuwang Wang. Incre-
> mental residual concept bottleneck models. In Proceedings of the IEEE Conference on Computer
> Vision and Pattern Recognition (CVPR), 2024.
> [4] Andong Tan, Fengtao Zhou, and Hao Chen. Explain via any concept: Concept bottleneck model
> with open vocabulary concepts. In Proceedings of the European Conference on Computer Vision
> (ECCV), 2024.
> [5] Chenyang Zhao, Kun Wang, Xingyu Zeng, Rui Zhao, and B. Antoni Chan. Gradient-based visual
> explanation for clip. In Proceedings of the International Conference on Machine Learning (ICML),
> 2024.

---

> ### Author Response · Authors · 2025-11-26
>
> ## Question 2 (part 1):
> We report the LLM-based interpretability evaluation across different models. In this evaluation, the LLM is asked to assess the quality of interpretations generated by our proposed CLIP-DMD and the baseline SpLiCE. Each interpretation includes the list of concepts shared between the image and caption, along with the corresponding sparse-feature similarity produced by each method. To mitigate positional bias (e.g., a tendency to favor the first option), we instruct the LLM to score each method's interpretation independently rather than choose between them. We additionally randomize the order in which the two methods are presented, to reduce ordering effects. After scoring all samples using this procedure, we compute how many times each method receives the highest score from the LLM to determine their relative interpretability. To account for randomness introduced by permutation and ordering, we repeat the evaluation three times and report the averaged results. The final results are presented in the table below:
>
> | Model | CLIP-DMD | SpLiCE |
> |:--------|:--------:|:--------:|
> | LLama-3.2-11B-Vision-Instruct| 0.6275 $\pm$  0.0243| 0.3725 $\pm$  0.0243
> | Gemma-3-27B-it| 0.7107 $\pm$  0.0007| 0.2893 $\pm$  0.0007|
> | Qwen2.5-VL-32B-Instruct| 0.5370 $\pm$  0.0032| 0.4630 $\pm$  0.0032|
>
> **The LLM evaluations with various models (including LLama, Gemma, and Qwen) show consistent results, indicating that our CLIP-DMD provides a more reasonable interpretation to reflect the similarity between image and caption by decomposed concepts, compared with SpLiCE.** We will also revise the section regarding the aforementioned LLM evaluation and the corresponding experimental results in our manuscript to ensure that the sufficient details are provided for improving readability.
>
> We also provide evidence of the alignment between LLM evaluation and human evaluation, demonstrating how closely the LLM is aligned with human decision-making. Specifically, we randomly sample 50 cases from the dataset and compare the preferences from LLMs and humans. The results are shown in the table below:
>
> | Model | Human Align rate |
> |:-----------|:----------------:|
> |LLama-3.2-11B-Vision-Instruct|0.59 $\pm$ 0.08|
> |Gemma-3-27-it|0.67 $\pm$ 0.02|
> |Qwen2.5-VL-32B-Instruct|0.59 $\pm$ 0.03|
>
> It is worth noting that the **Gemma archive shows the largest performance gap between CLIP-DMD and SpLiCE, indicating a stronger preference for the interpretations produced by CLIP-DMD**. Moreover, **Gemma has the highest human-alignment rate among the three evaluated models**, further reinforcing the reliability of the observed interpretability improvement.

---

> ### Author Response · Authors · 2025-11-26
>
> ## Question 2 (part 2):
> To enhance the reliability of the VLM evaluation, we further provide additional VLM evaluations by ensembling the decisions of different VLM models to assess which interpretation generated by CLIP-DMD or SpLiCE is more accurate. The various VLM models serve as different experts to discern which method provides better interpretations and vote, ultimately contributing to the final judgment. Specifically, we ensemble the decision from LLama-3.2-11B-Vision-Instruct, Gemma-3-27B-it, and Qwen2.5-VL-32B-Instruct. By providing the same prompt to these three VLM models, each model is asked to score the two provided interpretations (i.e. our CLIP-DMD and SpLiCE), and the highest score will indicate the model's preferred interpretation. We collect the decision from each model for each sample and use the voting process to decide which method the VLM models prefer more. We execute the process over the MS-COCO dataset three times and report the mean and standard deviation in the table below:
> |Model| CLIP-DMD | SpLiCE | Human Alignment rate |
> |:----------:|:--------:|:------:|:---------------:|
> | LLama-3.2-11B-Vision-Instruct + Gemma-3-27B-it + Qwen2.5-VL-32B-Instruct |  0.64 $\pm$ 0.0013  | 0.36 $\pm$ 0.0013 |    0.724 $\pm$ 0.0150   |
>
> To verify the degree of alignment between the ensemble VLM evaluation and human evaluation, we compare the decisions made by the ensemble VLM evaluation and human evaluation to present the alignment rate. Specifically, we randomly sample 50 samples and provide the image, corresponding caption, CLIP similarity, list of shared concepts between image and caption found by different methods, and the calculated similarity to users who have or are working towards a graduate degree in a STEM field and have knowledge of ML models, followed by asking them to select the most reasonable interpretation to explain the CLIP similarity by the provided concepts and similarities. We also adopt the ensemble VLM evaluation on the same set of samples and compare the decisions made by humans and VLM models to calculate the rate that reflects how aligned decisions are made by VLM and humans. The result presents the table above. **As the results indicate, the ensemble VLM evaluation consistently prefers the interpretations generated by CLIP-DMD, achieving a high degree of human alignment by integrating the decisions of three different models.** This finding mirrors the conclusion obtained when using individual LLMs, but provides a more reliable and robust assessment due to the ensemble’s aggregated judgment.
>
> ## Question 3:
> We thank the authors for pointing out the readability problem. We will carefully recheck that the revision contains sufficient detailed information.
>
> ## Question 4:
> We adopt the concept corpus introduced in SpLiCE, which is constructed by selecting the most frequent one-word and two-word bigrams from the text captions of the LAION-400M dataset. NSFW samples are removed, and concepts are further pruned to ensure that no two concepts have a similarity greater than 0.9. The final corpus consists of the 10000 most common single-word concepts and the 5000 most common two-word concepts.

---

> > ### Author Response · Authors · 2025-11-28
> >
> > We would greatly appreciate it if the reviewer could briefly confirm whether our responses and the newly added experiments have addressed your concerns. In the next few days, we will also update the revision paper. If there are any remaining questions or points that need further elaboration, we would be happy to address them promptly within the remaining discussion window. Thank you again for your time and thoughtful review. We sincerely value your feedback and engagement.

---

### Official Review · Reviewer_CQqb · 2025-10-30

**Soundness:** 2
**Presentation:** 2
**Contribution:** 2
**Rating:** 4
**Confidence:** 4

**Summary:**

This paper proposes CLIP-DMD, a sparse encoder–based framework designed to identify and interpret meaningful concepts embedded in CLIP features. Two novel loss functions are introduced to enhance the interpretability of the learned representations. However, the experimental results do not sufficiently validate the effectiveness of the proposed method, and its underlying motivation remains unclear. Moreover, figures and tables require substantial revision to improve clarity, consistency, and integration with the main text.

**Strengths:**

1. The proposed corpus cycle consistency loss and rate constraint loss contribute to the model’s ability to learn salient features.

2. The method is streamlined, contributing to implementation clarity.

**Weaknesses:**

1. Figure 1 appears abruptly and lacks corresponding explanation in the main text, making its relevance unclear. In addition, the performance drop of SpLiCE after introducing additional concepts is not addressed.

2. The authors argue that prior works either use CLIP as an auxiliary tool or focus on a single modality. However, methods like Grad-ECLIP already provide joint visual–textual explanations based on CLIP similarity. The authors should clarify what fundamentally distinguishes their approach, beyond simply citing CLIP usage or modality count.

3. Table 1 also appears abruptly without sufficient contextual grounding. The results presented do not clearly support the caption’s claim of being competitive on zero-shot classification, image-text retrieval, and linear probing benchmarks.

4. The architecture of the sparse text and image encoders is not clearly specified in the paper.

5. The proposed method relies on features extracted by CLIP. If these features are biased or flawed, the subsequent process may propagate such errors and compromise the model’s reliability.

6. The validity of the LLM evaluation process and the reliability of the reported results remain questionable.

7. The task agnosticity of concepts experiment lacks comparison with other baselines, which undermines its persuasiveness.

**Questions:**

1. The purpose of Figure 1 is unclear, and it does not illustrate how the interpretations are derived.

2. How are the values of the binary vector determined in the CCC loss?

3. Do all loss functions contribute equally during model training? According to Equation 6, all loss weights are set to 1.

4. The proposed method is mainly evaluated on recognition and classification tasks. It would be interesting to see its applicability to other tasks such as MLLM and image variation.

5. The paper does not analyze the effect of removing sparse text and image encoders, nor does it clarify the respective roles of CE and InfoNCE losses.

---

> ### Author Response · Authors · 2025-11-26
>
> We thank the reviewer for taking the time to provide valuable feedback. In the spirit of continually improving the quality of our work, we address each comment in turn, starting with the identified weaknesses and then responding to the specific questions.
>
> ## Weakness 1:
> The core idea of Figure 1 is to illustrate that the original CLIP model only outputs a similarity score between an image and a caption, without revealing how this similarity is derived. Although SpLiCE provides modality-specific concepts, the overlapping concepts and their computed similarity cannot explain why CLIP assigns a higher similarity to caption 2 than to caption 1, even yielding the opposite conclusion. In contrast, CLIP-DMD successfully figures out the core concept between the two pairs. By identifying the sea-related concept, our method explains why caption 2 achieves higher similarity to the image than caption 1. The similarity computed by considering these additional concepts aligns with CLIP’s behavior, correctly reflecting the higher similarity for caption 2.
>
> ## Weakness 2:
> The key distinction between Grad-ECLIP and our proposed CLIP-DMD lies in the depth of interpretability. **CLIP-DMD provides detailed insight into each concept’s contribution to the final similarity score, whereas Grad-ECLIP only highlights which concepts are present in the image and caption.** This major difference enables CLIP-DMD to provide interpretable sparse features for downstream tasks, such as linear probing, zero-shot classification, and retrieval; in contrast, the provided highlight interpretation from Grad-ECLIP cannot support such applications.
>
> ## Weakness 3 (part 1):
> We provide additional zero-shot evaluation results on Caltech101, CIFAR100, DTD, Flowers102, and PET, comparing our approach against CLIP and SpLiCE, as shown in the following table (we highlight in bold the highest-performing method among the interpretable approaches):
> |   Model  |  Method  | Interpretability|    Caltech101   |     CIFAR100    |       DTD       |     Flowers     |       PET       |
> |:--------:|:--------:|:--------:|:---------------:|:---------------:|:---------------:|:---------------:|:---------------:|
> | ResNet50 | CLIP|  |      79.86%     |      40.62%     |      38.56%     |      60.51%     |      83.21%     |
> |ResNet50|  SpLiCE | &check; |      0.33%      |      0.08%      |      0.05%      |      0.00%      |      0.05%      |
> |ResNet50| CLIP-DMD | &check;  | **82.57% $\pm$ 0.68%** | **39.75% $\pm$  0.25%** | **36.90% $\pm$ 0.66%** | **49.88% $\pm$ 1.18%** | **75.30% $\pm$ 0.29%** |
>
> |   Model  |  Method  | Interpretability|    Caltech101   |     CIFAR100    |       DTD       |     Flowers     |       PET       |
> |:--------:|:--------:|:--------:|:---------------:|:---------------:|:---------------:|:---------------:|:---------------:|
> | ViT-B-32 | CLIP| |      88.99%     |      75.91%     |      51.97%     |      73.23%     |      87.74%     |
> |ViT-B-32|  SpLiCE| &check; |      0.47%      |      0.39%      |      0.27%      |      0.49%      |      0.30%      |
> |ViT-B-32| CLIP-DMD | &check; | **90.12%  $\pm$  0.48%** | **72.87%  $\pm$  0.30%** | **49.13%  $\pm$  0.98%** | **64.75%  $\pm$  1.21%** | **83.89%  $\pm$  0.78%** |
>
> |   Model  |  Method  | Interpretability|    Caltech101   |     CIFAR100    |       DTD       |     Flowers     |       PET       |
> |:--------:|:--------:|:--------:|:---------------:|:---------------:|:---------------:|:---------------:|:---------------:|
> | ViT-B-16 | CLIP | |      89.22%     |      77.23%     |      52.98%     |      71.15%     |      89.45%     |
> |ViT-B-16|  SpLiCE | &check;|      0.52%      |      0.08%      |      0.05%      |      0.00%      |      0.19%      |
> |ViT-B-16| CLIP-DMD | &check; | **88.16%  $\pm$  0.73%** | **75.05%  $\pm$  0.27%** | **48.67%  $\pm$  1.87%** | **66.18%  $\pm$  0.90%** | **84.15%  $\pm$  0.10%** |
>
> Although CLIP-DMD exhibits some performance degradation relative to CLIP, it offers deeper insight into similarity by revealing why certain similarities occur. **When compared to SpLiCE, another interpretable method that decomposes different modalities separately without considering their relation, CLIP-DMD consistently outperforms it.**

---

> ### Author Response · Authors · 2025-11-26
>
> ## Weakness 3 (part 2):
> We also conduct linear probing experiments to demonstrate that the sparse features learned by CLIP-DMD remain effective for downstream tasks. For each experiment, we train a classifier consisting of a single fully connected layer with bias for 100 epochs using AdamW. The learning rate is set to $10^{-2}$ for all datasets. We highlight in bold the highest-performing method among the interpretable approaches
>
> | Backbone |  Method | Interpretability |   Caltech101  |    CIFAR100   |      DTD      |   Flowers102  |      PET      |
> |:--------:|:--------:|:--------:|:-------------:|:-------------:|:-------------:|:-------------:|:-------------:|
> | ResNet50 | CLIP| | 94.56% $\pm$ 0.03% | 69.89% $\pm$ 0.06% | 72.91% $\pm$ 0.17% | 97.80% $\pm$ 0.00% | 87.53% $\pm$ 0.04% |
> |ResNet50|  SpLiCE | &check; | 37.00% $\pm$ 0.10% | 42.68% $\pm$ 0.06% | 57.96% $\pm$ 0.03% | 23.15% $\pm$ 1.19% | 52.04% $\pm$ 0.06% |
> |ResNet50|  DN-CBM| &check;| **93.54% $\pm$ 0.10%** | 68.72% $\pm$ 0.06% | 70.41% $\pm$ 0.08% | 95.64% $\pm$ 0.07% | 77.59% $\pm$ 0.03% |
> |ResNet50| CLIP-DMD | &check; | 93.41% $\pm$ 0.05% | **69.87% $\pm$ 0.03%** | **71.99% $\pm$ 0.12%** | **97.31% $\pm$ 0.21%** | **86.37% $\pm$ 0.00%** |
>
> | Backbone |  Method | Interpretability |   Caltech101  |    CIFAR100   |      DTD      |   Flowers102  |      PET      |
> |:--------:|:--------:|:--------:|:-------------:|:-------------:|:-------------:|:-------------:|:-------------:|
> | ViT-B-32 | CLIP | | 97.60% $\pm$ 0.17% | 85.75% $\pm$ 0.06% | 79.26% $\pm$ 0.21% | 98.66% $\pm$ 0.00% | 90.63% $\pm$ 0.07% |
> |ViT-B-32|  SpLiCE| &check;| 39.37% $\pm$ 0.18% | 67.50% $\pm$ 0.05% | 63.23% $\pm$ 0.03% | 33.62% $\pm$ 2.12% | 59.78% $\pm$ 0.06% |
> |ViT-B-32|  DN-CBM | &check;| 97.52% $\pm$ 0.03% | 86.00% $\pm$ 0.06% | 79.22% $\pm$ 0.11% | 98.17% $\pm$ 0.00% | 88.87% $\pm$ 0.04% |
> |ViT-B-32| CLIP-DMD | &check; | **97.58% $\pm$ 0.05%** | **86.36% $\pm$ 0.02%** | **79.73% $\pm$ 0.05%** | **98.21% $\pm$ 0.07%** | **90.21% $\pm$ 0.08%** |
>
> | Backbone |  Method | Interpretability |   Caltech101  |    CIFAR100   |      DTD      |   Flowers102  |      PET      |
> |:--------:|:--------:|:--------:|:-------------:|:-------------:|:-------------:|:-------------:|:-------------:|
> | ViT-B-16 | CLIP | | 97.77% $\pm$ 0.12% | 86.13% $\pm$ 0.13% | 81.65% $\pm$ 0.09% | 99.10% $\pm$ 0.07% | 92.26% $\pm$ 0.14% |
> |ViT-B-16|  SpLiCE | &check; | 39.50% $\pm$ 0.23% | 71.03% $\pm$ 0.04% | 64.66% $\pm$ 0.06% | 33.99% $\pm$ 1.48% | 65.40% $\pm$ 0.14% |
> |ViT-B-16|  DN-CBM | &check; | **97.74% $\pm$ 0.12%** | 86.53% $\pm$ 0.09% | 81.19% $\pm$ 0.08% | 98.82% $\pm$ 0.07% | 90.87% $\pm$ 0.03% |
> |ViT-B-16| CLIP-DMD | &check; | 97.69% $\pm$ 0.05% | **86.58% $\pm$ 0.07%** | **81.56% $\pm$ 0.11%** | **98.90% $\pm$ 0.00%** | **92.82% $\pm$ 0.02%** |
>
> **The results show that CLIP-DMD achieves performance comparable to CLIP, a non-interpretable method, across various models and datasets**. In contrast, DN-CBM exhibits performance drops in some settings (DTD, Flowers102 on ResNet50, and all models on PET), and SpLiCE fails to match the performance of either the CLIP or CLIP-DMD.
>
> ## Weakness 4:
> The sparse text and image encoders and decoders are implemented as fully connected layers without bias terms. For the sparse encoders, we further include a thresholding layer after the linear transformation, which functions similarly to a ReLU activation when the threshold is set to 0, used to prevent negative concept weights.
>
> ## Weakness 5:
> The core objective of CLIP-DMD is to provide insight into the CLIP similarity by presenting the concepts shared between an image and its corresponding caption, along with the calculated similarity, which exhibits a similar trend to the CLIP similarity. **The biases and imperfections revealed by CLIP-DMD can be viewed as a strength, as they help illuminate CLIP’s limitations and may guide future improvements in its feature representations.**

---

> ### Author Response · Authors · 2025-11-26
>
> ## Weakness 6 (part 1):
> We report the LLM-based interpretability evaluation across different models. In this evaluation, the LLM is asked to assess the quality of interpretations generated by our proposed CLIP-DMD and the baseline SpLiCE. Each interpretation includes the list of concepts shared between the image and caption, along with the corresponding sparse-feature similarity produced by each method. To mitigate positional bias (e.g., a tendency to favor the first option), we instruct the LLM to score each method's interpretation independently rather than choose between them. We additionally randomize the order in which the two methods are presented, to reduce ordering effects. After scoring all samples using this procedure, we compute how many times each method receives the highest score from the LLM to determine their relative interpretability. To account for randomness introduced by permutation and ordering, we repeat the evaluation three times and report the averaged results. The final results are presented in the table below:
>
> | Model | CLIP-DMD | SpLiCE |
> |:--------|:--------:|:--------:|
> | LLama-3.2-11B-Vision-Instruct| 0.6275 $\pm$  0.0243| 0.3725 $\pm$  0.0243
> | Gemma-3-27B-it| 0.7107 $\pm$  0.0007| 0.2893 $\pm$  0.0007|
> | Qwen2.5-VL-32B-Instruct| 0.5370 $\pm$  0.0032| 0.4630 $\pm$  0.0032|
>
> **The LLM evaluations with various models (including LLama, Gemma, and Qwen) show consistent results, indicating that our CLIP-DMD provides a more reasonable interpretation to reflect the similarity between image and caption by decomposed concepts, compared with SpLiCE.** We will also revise the section regarding the aforementioned LLM evaluation and the corresponding experimental results in our manuscript to ensure that the sufficient details are provided for improving readability.
>
> We also provide evidence of the alignment between LLM evaluation and human evaluation, demonstrating how closely the LLM is aligned with human decision-making. Specifically, we randomly sample 50 cases from the dataset and compare the preferences from LLMs and humans. The results are shown in the table below:
>
> | Model | Human Align rate |
> |:-----------|:----------------:|
> |LLama-3.2-11B-Vision-Instruct|0.59 $\pm$ 0.08|
> |Gemma-3-27-it|0.67 $\pm$ 0.02|
> |Qwen2.5-VL-32B-Instruct|0.59 $\pm$ 0.03|
>
> It is worth noting that the **Gemma archive shows the largest performance gap between CLIP-DMD and SpLiCE, indicating a stronger preference for the interpretations produced by CLIP-DMD**. Moreover, **Gemma has the highest human-alignment rate among the three evaluated models**, further reinforcing the reliability of the observed interpretability improvement.

---

> ### Author Response · Authors · 2025-11-26
>
> ## Weakness 6 (part 2):
> To enhance the reliability of the VLM evaluation, we further provide additional VLM evaluations by ensembling the decisions of different VLM models to assess which interpretation generated by CLIP-DMD or SpLiCE is more accurate. The various VLM models serve as different experts to discern which method provides better interpretations and vote, ultimately contributing to the final judgment. Specifically, we ensemble the decision from LLama-3.2-11B-Vision-Instruct, Gemma-3-27B-it, and Qwen2.5-VL-32B-Instruct. By providing the same prompt to these three VLM models, each model is asked to score the two provided interpretations (i.e. our CLIP-DMD and SpLiCE), and the highest score will indicate the model's preferred interpretation. We collect the decision from each model for each sample and use the voting process to decide which method the VLM models prefer more. We execute the process over the MS-COCO dataset three times and report the mean and standard deviation in the table below:
> |Model| CLIP-DMD | SpLiCE | Human Alignment rate |
> |:----------:|:--------:|:------:|:---------------:|
> | LLama-3.2-11B-Vision-Instruct + Gemma-3-27B-it + Qwen2.5-VL-32B-Instruct |  0.64 $\pm$ 0.0013  | 0.36 $\pm$ 0.0013 |    0.724 $\pm$ 0.0150   |
>
> To verify the degree of alignment between the ensemble VLM evaluation and human evaluation, we compare the decisions made by the ensemble VLM evaluation and human evaluation to present the alignment rate. Specifically, we randomly sample 50 samples and provide the image, corresponding caption, CLIP similarity, list of shared concepts between image and caption found by different methods, and the calculated similarity to users who have or are working towards a graduate degree in a STEM field and have knowledge of ML models, followed by asking them to select the most reasonable interpretation to explain the CLIP similarity by the provided concepts and similarities. We also adopt the ensemble VLM evaluation on the same set of samples and compare the decisions made by humans and VLM models to calculate the rate that reflects how aligned decisions are made by VLM and humans. The result presents the table above. **As the results indicate, the ensemble VLM evaluation consistently prefers the interpretations generated by CLIP-DMD, achieving a high degree of human alignment by integrating the decisions of three different models.** This finding mirrors the conclusion obtained when using individual LLMs, but provides a more reliable and robust assessment due to the ensemble’s aggregated judgment.
>
> ## Weakness 7:
> The strong performance on zero-shot, retrieval, and linear probing tasks demonstrates that CLIP-DMD produces useful and practical sparse features that can be applied to a variety of downstream tasks without requiring any retraining of CLIP-DMD.
>
> ## Question 1:
> As reply in [**Weakness 1**], the core idea of Figure 1 is to illustrate that the original CLIP model only outputs a similarity score between an image and a caption, without revealing how this similarity is derived. Although SpLiCE provides modality-specific concepts, the overlapping concepts and their computed similarity cannot explain why CLIP assigns a higher similarity to caption 2 than to caption 1, even yielding the opposite conclusion. In contrast, CLIP-DMD successfully figures out the core concept between the two pairs. By identifying the sea-related concept, our method explains why caption 2 achieves higher similarity to the image than caption 1. The similarity computed by considering these additional concepts aligns with CLIP’s behavior, correctly reflecting the higher similarity for caption 2.
>
> ## Question 2:
> The binary vector for the CCC loss is determined by letting the same place label be 1 and 0 for the rest of the part. Specifically, the corpus features $W^{corpus} \in \mathbb{R}^{D \times M}$ in the decoder can be treated as having M D-dimensional corpus features representing concepts. Each $w_i \in W^{corpus}$ will have the corresponding binary label with i-th dimension set to be 1 and the rest of the parts are 0 as the ground truth during calculating the CCC loss. This forces the text encoder to learn correctly reflecting the corpus features by making the corresponding place (e.g. i-th) the greatest.
>
> ## Question 3:
> All loss terms are assigned equal weight during training; specifically, each loss coefficient is set to 1.
>
> ## Question 4:
> Our CLIP-DMD framework is designed to provide deeper insight into CLIP’s similarity computation by revealing the underlying concept composition that contributes to the score. The applications mentioned by the reviewer fall outside the scope of our current focus, but might be an interesting application for future work.

---

> ### Author Response · Authors · 2025-11-26
>
> ## Question 5:
> **The sparse text and image encoders are core components of our CLIP-DMD framework.** As part of the SAE, they are responsible for learning the concept composition. Removing either the encoders or the decoder would essentially remove our method itself, rendering the proposed losses inapplicable. The CE loss is used for the CCC loss, which forces the corresponding dimension of the feature to have the greater value than the rest. The purpose of the InfoNCE loss pulls the pairs of image and text representation closer from the learned sparse feature space and enlarges the distance between the non-pairs.

---

> > ### Author Response · Authors · 2025-11-28
> >
> > We would greatly appreciate it if the reviewer could briefly confirm whether our responses and the newly added experiments have addressed your concerns. In the next few days, we will also update the revision paper. If there are any remaining questions or points that need further elaboration, we would be happy to address them promptly within the remaining discussion window. Thank you again for your time and thoughtful review. We sincerely value your feedback and engagement.

---

### Official Review · Reviewer_7TfT · 2025-10-30

**Soundness:** 3
**Presentation:** 3
**Contribution:** 3
**Rating:** 4
**Confidence:** 4

**Summary:**

This paper addresses the limited interpretability of CLIP’s cross-modal similarity, which is typically measured via cosine similarity in a high-dimensional space but lacks insight into which semantic concepts drive this similarity. The authors propose CLIP-DMD, a method that uses a Sparse Autoencoder (SAE) to decompose both image and text CLIP features into a shared, sparse concept space. Key innovations include two novel losses: the Corpus Cycle Consistency (C³) loss, which ensures that learned concepts are accurately recognized by the encoder, and the Rate Constraint (RC) loss, which encourages similarity to be dominated by a small set of salient concepts. The authors also introduce an LLM-based evaluation protocol to automatically assess interpretability. Experiments show that CLIP-DMD maintains competitive performance on zero-shot classification, retrieval, and linear probing tasks while providing more human-understandable explanations than prior methods like SpLiCE.

**Strengths:**

1. This paper introduces a unified, dual-modality sparse decomposition framework that enables interpretable cross-modal similarity analysis.
2. Authors propose two novel losses (C³ and RC) that enhance concept recognition and saliency, supported by ablation studies.

**Weaknesses:**

1. The predefined unigram/bigram corpus limits the granularity and expressiveness of discovered concepts, as noted in the limitations.
2. LLM-based interpretability evaluation is not validated against human judgments, raising questions about its reliability.
3. The method does not fully address how to handle compositional or abstract concepts beyond the corpus vocabulary.

**Questions:**

1. How does the LLM-based interpretability score correlate with human judgments? Have the authors conducted any human evaluation to validate this metric?
2. Could the authors explore more expressive concept naming strategies (e.g., using phrases or LLM-generated descriptions) to overcome the limitations of unigrams/bigrams?
3. How does CLIP-DMD handle cases where multiple concepts interact or overlap semantically? Is there a risk of over-simplification when only top-k concepts are highlighted?
4. Could the author provide more intuition or visualization for how C³ loss enables encoders to accurately respond to the learned corpus features?

---

> ### Author Response · Authors · 2025-11-26
>
> We thank the reviewer for taking the time to provide valuable feedback. In the spirit of continually improving the quality of our work, we address each comment in turn, starting with the identified weaknesses and then responding to the specific questions.
>
> ## Weakness 1:
> This restriction stems from the design of the original approach, which learned and then named the features afterward; previous works also suffer from the same limitation, e.g., SpLiCE and DN-CBM. Since our contribution lies in learning meaningful sparse features to represent the weights of concepts that exist in the image, and the image provides deeper insight into the similarity, we select and retain the original corpus unchanged. The method that names the feature without the predefined corpus could be an important direction for future work.
>
> ## Weakness 2 (part 1):
> We report the LLM-based interpretability evaluation across different models. In this evaluation, the LLM is asked to assess the quality of interpretations generated by our proposed CLIP-DMD and the baseline SpLiCE. Each interpretation includes the list of concepts shared between the image and caption, along with the corresponding sparse-feature similarity produced by each method. To mitigate positional bias (e.g., a tendency to favor the first option), we instruct the LLM to score each method's interpretation independently rather than choose between them. We additionally randomize the order in which the two methods are presented, to reduce ordering effects. After scoring all samples using this procedure, we compute how many times each method receives the highest score from the LLM to determine their relative interpretability. To account for randomness introduced by permutation and ordering, we repeat the evaluation three times and report the averaged results. The final results are presented in the table below:
>
> | Model | CLIP-DMD | SpLiCE |
> |:--------|:--------:|:--------:|
> | LLama-3.2-11B-Vision-Instruct| 0.6275 $\pm$  0.0243| 0.3725 $\pm$  0.0243
> | Gemma-3-27B-it| 0.7107 $\pm$  0.0007| 0.2893 $\pm$  0.0007|
> | Qwen2.5-VL-32B-Instruct| 0.5370 $\pm$  0.0032| 0.4630 $\pm$  0.0032|
>
> **The LLM evaluations with various models (including LLama, Gemma, and Qwen) show consistent results, indicating that our CLIP-DMD provides a more reasonable interpretation to reflect the similarity between image and caption by decomposed concepts, compared with SpLiCE.** We will also revise the section regarding the aforementioned LLM evaluation and the corresponding experimental results in our manuscript to ensure that the sufficient details are provided for improving readability.
>
> We also provide evidence of the alignment between LLM evaluation and human evaluation, demonstrating how closely the LLM is aligned with human decision-making. Specifically, we randomly sample 50 cases from the dataset and compare the preferences from LLMs and humans. The results are shown in the table below:
>
> | Model | Human Align rate |
> |:-----------|:----------------:|
> |LLama-3.2-11B-Vision-Instruct|0.59 $\pm$ 0.08|
> |Gemma-3-27-it|0.67 $\pm$ 0.02|
> |Qwen2.5-VL-32B-Instruct|0.59 $\pm$ 0.03|
>
> It is worth noting that the **Gemma archive shows the largest performance gap between CLIP-DMD and SpLiCE, indicating a stronger preference for the interpretations produced by CLIP-DMD**. Moreover, **Gemma has the highest human-alignment rate among the three evaluated models**, further reinforcing the reliability of the observed interpretability improvement.

---

> ### Author Response · Authors · 2025-11-26
>
> ## Weakness 2 (part 2):
> To enhance the reliability of the VLM evaluation, we further provide additional VLM evaluations by ensembling the decisions of different VLM models to assess which interpretation generated by CLIP-DMD or SpLiCE is more accurate. The various VLM models serve as different experts to discern which method provides better interpretations and vote, ultimately contributing to the final judgment. Specifically, we ensemble the decision from LLama-3.2-11B-Vision-Instruct, Gemma-3-27B-it, and Qwen2.5-VL-32B-Instruct. By providing the same prompt to these three VLM models, each model is asked to score the two provided interpretations (i.e. our CLIP-DMD and SpLiCE), and the highest score will indicate the model's preferred interpretation. We collect the decision from each model for each sample and use the voting process to decide which method the VLM models prefer more. We execute the process over the MS-COCO dataset three times and report the mean and standard deviation in the table below:
> |Model| CLIP-DMD | SpLiCE | Human Alignment rate |
> |:----------:|:--------:|:------:|:---------------:|
> | LLama-3.2-11B-Vision-Instruct + Gemma-3-27B-it + Qwen2.5-VL-32B-Instruct |  0.64 $\pm$ 0.0013  | 0.36 $\pm$ 0.0013 |    0.724 $\pm$ 0.0150   |
>
> To verify the degree of alignment between the ensemble VLM evaluation and human evaluation, we compare the decisions made by the ensemble VLM evaluation and human evaluation to present the alignment rate. Specifically, we randomly sample 50 samples and provide the image, corresponding caption, CLIP similarity, list of shared concepts between image and caption found by different methods, and the calculated similarity to users who have or are working towards a graduate degree in a STEM field and have knowledge of ML models, followed by asking them to select the most reasonable interpretation to explain the CLIP similarity by the provided concepts and similarities. We also adopt the ensemble VLM evaluation on the same set of samples and compare the decisions made by humans and VLM models to calculate the rate that reflects how aligned decisions are made by VLM and humans. The result presents the table above. **As the results indicate, the ensemble VLM evaluation consistently prefers the interpretations generated by CLIP-DMD, achieving a high degree of human alignment by integrating the decisions of three different models.** This finding mirrors the conclusion obtained when using individual LLMs, but provides a more reliable and robust assessment due to the ensemble’s aggregated judgment.
>
> ## Weakness 3:
> Our method adopts the concept naming procedure used in DN-CBM, which inherently shares the same limitation regarding the fixed corpus size; the same applies to SpLiCE. As noted in [**Weakness 1**], exploring approaches that learn concepts without relying on a predefined corpus for naming represents a promising future direction.
>
> ## Question 1:
> As addressed in [**Weakness 2**], we provide additional LLM-based evaluations across multiple models, which consistently support our findings, along with an alignment analysis between the LLMs and human judgments.
> We did not directly rely on human evaluation, as it typically involves only a small number of samples, whereas LLM-based evaluation can be applied to the entire dataset without requiring extensive manual effort.
>
> ## Question 2:
> We provide a potential alternative for naming the learned concepts. One possibility is to collect the top-k response images from a large-scale dataset (e.g., ImageNet or MS-COCO) and submit them to an LLM to generate concise descriptions that capture the shared semantic meaning of each concept. However, this strategy cannot guarantee providing a consistent concept set across different methods, which would result in misaligned concept vocabularies and lead to unfair comparisons of interpretability.
> While exploring a more flexible naming mechanism that removes the reliance on a fixed, limited corpus is an interesting direction for future work, our current focus is on developing discriminative and interpretable sparse features and learning meaningful corpus representations that offer deeper insight into CLIP’s similarity behavior.
>
> ## Question 3:
> We present only the top-k concepts because they represent the most salient or influential concepts detected in the image or caption. More sophisticated selection strategies remain possible and can be explored in future work. In this paper, we simply adopt top-k selection strategy for the fair comparison between various methods.

---

> > ### Author Response · Authors · 2025-11-26
> >
> > ## Question 4:
> > The corpus cycle consistency loss enforces that the learned corpus weights $W^{corpus}\in \mathbb{R}^{D \times M}$ , the decoder parameters shared across both modalities to define a unified concept space, should also be correctly recognized by the text encoder $Enc_{text}(.)$. These corpus weights are trained to reconstruct CLIP features, using weights predicted by the text encoder when a CLIP feature is provided as input.
> > More concretely, each corpus vector $w_i \in W^{corpus}$ corresponds to a specific semantic concept and is treated as the input to the text encoder. Ideally, the sparse output of the text encoder should activate only the i-th dimension (i.e., produce a 1 at position i and 0 elsewhere). This ensures that, after passing through the decoder, only the i-th row of the learned corpus weights is used to reconstruct the feature, reflecting the meaning represented by $w_i$.
> > Intuitively, this relationship can be expressed as: $w_i=Dec(Enc_{text}(w_i))$.

---

### Official Review · Reviewer_sH98 · 2025-10-31

**Soundness:** 3
**Presentation:** 3
**Contribution:** 2
**Rating:** 4
**Confidence:** 4

**Summary:**

The problem of interpretable CLIP is considered.  The paper introduces CLIP-DMD (CLIP Similarity via Dual Modalities Decomposition), a novel framework that explains the similarity between image and text features in the CLIP model by decomposing them into a shared, sparse concept space using a Sparse Autoencoder (SAE). To enhance interpretability, CLIP-DMD uses two novel objectives: the Rate Constraint (RC) Loss, which ensures that the similarity is dominated by a few critical concepts, and the Corpus Cycle Consistency ($C^3$) Loss, which improves the encoder's recognition of the learned concepts. Quantitative and qualitative results show that CLIP-DMD maintains competitive performance on tasks like zero-shot classification and retrieval while offering superior interpretability and a closer reflection of CLIP's similarity trends compared to previous methods.

**Strengths:**

1. Technical novelties: Corpus Cycle Consistency (C^3) and Rate Constraint (RC) losses
2. Empirical results validating the effectiveness of the proposed method.

**Weaknesses:**

1. The interpretability evaluation is merely based on a single multi-modal LLM LLaMA-3.2-Vision-11B-Instruction. The current mLLMs are known to can be biased and can make mistakes. The trust-worthiness of the empirical study remains to be questioned. Additional Evaluations by more different mLLMs and human experts are necessary.
2. Lack of 0-shot evaluation on more datasets. The common practice to evaluate the 0-shot performance of CLIP models is to evaluate on a variety of different datasets (e.g. Pets, Foods, Sun, etc, see VTAB+ for more). Also, why  is the linear probing evaluation only on CUB and TinyImagenet?
3. Lack of demonstration of downstream application.

**Questions:**

N/A

---

> ### Author Response · Authors · 2025-11-26
>
> We thank the reviewer for taking the time to provide valuable feedback. In the spirit of improving the quality of our work, we address the identified weaknesses.
>
> ## Weakness 1 (part 1):
> We report the LLM-based interpretability evaluation across different models. In this evaluation, the LLM is asked to assess the quality of interpretations generated by our proposed CLIP-DMD and the baseline SpLiCE. Each interpretation includes the list of concepts shared between the image and caption, along with the corresponding sparse-feature similarity produced by each method. To mitigate positional bias (e.g., a tendency to favor the first option), we instruct the LLM to score each method's interpretation independently rather than choose between them. We additionally randomize the order in which the two methods are presented, to reduce ordering effects. After scoring all samples using this procedure, we compute how many times each method receives the highest score from the LLM to determine their relative interpretability. To account for randomness introduced by permutation and ordering, we repeat the evaluation three times and report the averaged results. The final results are presented in the table below:
>
> | Model | CLIP-DMD | SpLiCE |
> |:--------|:--------:|:--------:|
> | LLama-3.2-11B-Vision-Instruct| 0.6275 $\pm$  0.0243| 0.3725 $\pm$  0.0243
> | Gemma-3-27B-it| 0.7107 $\pm$  0.0007| 0.2893 $\pm$  0.0007|
> | Qwen2.5-VL-32B-Instruct| 0.5370 $\pm$  0.0032| 0.4630 $\pm$  0.0032|
>
> **The LLM evaluations with various models (including LLama, Gemma, and Qwen) show consistent results, indicating that our CLIP-DMD provides a more reasonable interpretation to reflect the similarity between image and caption by decomposed concepts, compared with SpLiCE.** We will also revise the section regarding the aforementioned LLM evaluation and the corresponding experimental results in our manuscript to ensure that the sufficient details are provided for improving readability.
>
> We also provide evidence of the alignment between LLM evaluation and human evaluation, demonstrating how closely the LLM is aligned with human decision-making. Specifically, we randomly sample 50 cases from the dataset and compare the preferences from LLMs and humans. The results are shown in the table below:
>
> | Model | Human Align rate |
> |:-----------|:----------------:|
> |LLama-3.2-11B-Vision-Instruct|0.59 $\pm$ 0.08|
> |Gemma-3-27-it|0.67 $\pm$ 0.02|
> |Qwen2.5-VL-32B-Instruct|0.59 $\pm$ 0.03|
>
> It is worth noting that the **Gemma archive shows the largest performance gap between CLIP-DMD and SpLiCE, indicating a stronger preference for the interpretations produced by CLIP-DMD.** Moreover, **Gemma has the highest human-alignment rate among the three evaluated models**, further reinforcing the reliability of the observed interpretability improvement.

---

> ### Author Response · Authors · 2025-11-26
>
> ## Weakness 1 (part 2):
> To enhance the reliability of the VLM evaluation, we further provide additional VLM evaluations by ensembling the decisions of different VLM models to assess which interpretation generated by CLIP-DMD or SpLiCE is more accurate. The various VLM models serve as different experts to discern which method provides better interpretations and vote, ultimately contributing to the final judgment. Specifically, we ensemble the decision from LLama-3.2-11B-Vision-Instruct, Gemma-3-27B-it, and Qwen2.5-VL-32B-Instruct. By providing the same prompt to these three VLM models, each model is asked to score the two provided interpretations (i.e. our CLIP-DMD and SpLiCE), and the highest score will indicate the model's preferred interpretation. We collect the decision from each model for each sample and use the voting process to decide which method the VLM models prefer more. We execute the process over the MS-COCO dataset three times and report the mean and standard deviation in the table below:
> |Model| CLIP-DMD | SpLiCE | Human Alignment rate |
> |:----------:|:--------:|:------:|:---------------:|
> | LLama-3.2-11B-Vision-Instruct + Gemma-3-27B-it + Qwen2.5-VL-32B-Instruct |  0.64 $\pm$ 0.0013  | 0.36 $\pm$ 0.0013 |    0.724 $\pm$ 0.0150   |
>
> To verify the degree of alignment between the ensemble VLM evaluation and human evaluation, we compare the decisions made by the ensemble VLM evaluation and human evaluation to present the alignment rate. Specifically, we randomly sample 50 samples and provide the image, corresponding caption, CLIP similarity, list of shared concepts between image and caption found by different methods, and the calculated similarity to users who have or are working towards a graduate degree in a STEM field and have knowledge of ML models, followed by asking them to select the most reasonable interpretation to explain the CLIP similarity by the provided concepts and similarities. We also adopt the ensemble VLM evaluation on the same set of samples and compare the decisions made by humans and VLM models to calculate the rate that reflects how aligned decisions are made by VLM and humans. The result presents the table above. **As the results indicate, the ensemble VLM evaluation consistently prefers the interpretations generated by CLIP-DMD, achieving a high degree of human alignment by integrating the decisions of three different models.** This finding mirrors the conclusion obtained when using individual LLMs, but provides a more reliable and robust assessment due to the ensemble’s aggregated judgment.

---

> ### Author Response · Authors · 2025-11-26
>
> ## Weakness 2 (part 1):
> We provide additional zero-shot evaluation results on Caltech101, CIFAR100, DTD, Flowers102, and PET, comparing our approach against CLIP and SpLiCE, as shown in the following table (we highlight in bold the highest-performing method among the interpretable approaches):
> |   Model  |  Method  | Interpretability|    Caltech101   |     CIFAR100    |       DTD       |     Flowers     |       PET       |
> |:--------:|:--------:|:--------:|:---------------:|:---------------:|:---------------:|:---------------:|:---------------:|
> | ResNet50 | CLIP|  |      79.86%     |      40.62%     |      38.56%     |      60.51%     |      83.21%     |
> |ResNet50|  SpLiCE | &check; |      0.33%      |      0.08%      |      0.05%      |      0.00%      |      0.05%      |
> |ResNet50| CLIP-DMD | &check;  | **82.57% $\pm$ 0.68%** | **39.75% $\pm$  0.25%** | **36.90% $\pm$ 0.66%** | **49.88% $\pm$ 1.18%** | **75.30% $\pm$ 0.29%** |
>
> |   Model  |  Method  | Interpretability|    Caltech101   |     CIFAR100    |       DTD       |     Flowers     |       PET       |
> |:--------:|:--------:|:--------:|:---------------:|:---------------:|:---------------:|:---------------:|:---------------:|
> | ViT-B-32 | CLIP|  |      88.99%     |      75.91%     |      51.97%     |      73.23%     |      87.74%     |
> |ViT-B-32|  SpLiCE| &check; |      0.47%      |      0.39%      |      0.27%      |      0.49%      |      0.30%      |
> |ViT-B-32| CLIP-DMD | &check; | **90.12%  $\pm$  0.48%** | **72.87%  $\pm$  0.30%** | **49.13%  $\pm$  0.98%** | **64.75%  $\pm$  1.21%** | **83.89%  $\pm$  0.78%** |
>
> |   Model  |  Method  | Interpretability|    Caltech101   |     CIFAR100    |       DTD       |     Flowers     |       PET       |
> |:--------:|:--------:|:--------:|:---------------:|:---------------:|:---------------:|:---------------:|:---------------:|
> | ViT-B-16 | CLIP | |      89.22%     |      77.23%     |      52.98%     |      71.15%     |      89.45%     |
> |ViT-B-16|  SpLiCE | &check;|      0.52%      |      0.08%      |      0.05%      |      0.00%      |      0.19%      |
> |ViT-B-16| CLIP-DMD | &check; | **88.16%  $\pm$  0.73%** | **75.05%  $\pm$  0.27%** | **48.67%  $\pm$  1.87%** | **66.18%  $\pm$  0.90%** | **84.15%  $\pm$  0.10%** |
>
> Although CLIP-DMD exhibits some performance degradation relative to CLIP, it offers deeper insight into similarity by revealing why certain similarities occur. **When compared to SpLiCE, another interpretable method that decomposes different modalities separately without considering their relation, CLIP-DMD consistently outperforms it.**

---

> ### Author Response · Authors · 2025-12-03
>
> ## Weakness 2 (part 2):
> We also conduct linear probing experiments to demonstrate that the sparse features learned by CLIP-DMD remain effective for downstream tasks. For each experiment, we train a classifier consisting of a single fully connected layer with bias for 100 epochs using AdamW. The learning rate is set to $10^{-2}$ for all datasets. We highlight in bold the highest-performing method among the interpretable approaches
>
> | Backbone |  Method | Interpretability |   Caltech101  |    CIFAR100   |      DTD      |   Flowers102  |      PET      |
> |:--------:|:--------:|:--------:|:-------------:|:-------------:|:-------------:|:-------------:|:-------------:|
> | ResNet50 | CLIP| | 94.56% $\pm$ 0.03% | 69.89% $\pm$ 0.06% | 72.91% $\pm$ 0.17% | 97.80% $\pm$ 0.00% | 87.53% $\pm$ 0.04% |
> |ResNet50|  SpLiCE | &check; | 37.00% $\pm$ 0.10% | 42.68% $\pm$ 0.06% | 57.96% $\pm$ 0.03% | 23.15% $\pm$ 1.19% | 52.04% $\pm$ 0.06% |
> |ResNet50|  DN-CBM| &check;| **93.54% $\pm$ 0.10%** | 68.72% $\pm$ 0.06% | 70.41% $\pm$ 0.08% | 95.64% $\pm$ 0.07% | 77.59% $\pm$ 0.03% |
> |ResNet50| CLIP-DMD | &check; | 93.41% $\pm$ 0.05% | **69.87% $\pm$ 0.03%** | **71.99% $\pm$ 0.12%** | **97.31% $\pm$ 0.21%** | **86.37% $\pm$ 0.00%** |
>
> | Backbone |  Method | Interpretability |   Caltech101  |    CIFAR100   |      DTD      |   Flowers102  |      PET      |
> |:--------:|:--------:|:--------:|:-------------:|:-------------:|:-------------:|:-------------:|:-------------:|
> | ViT-B-32 | CLIP | | 97.60% $\pm$ 0.17% | 85.75% $\pm$ 0.06% | 79.26% $\pm$ 0.21% | 98.66% $\pm$ 0.00% | 90.63% $\pm$ 0.07% |
> |ViT-B-32|  SpLiCE| &check;| 39.37% $\pm$ 0.18% | 67.50% $\pm$ 0.05% | 63.23% $\pm$ 0.03% | 33.62% $\pm$ 2.12% | 59.78% $\pm$ 0.06% |
> |ViT-B-32|  DN-CBM | &check;| 97.52% $\pm$ 0.03% | 86.00% $\pm$ 0.06% | 79.22% $\pm$ 0.11% | 98.17% $\pm$ 0.00% | 88.87% $\pm$ 0.04% |
> |ViT-B-32| CLIP-DMD | &check; | **97.58% $\pm$ 0.05%** | **86.36% $\pm$ 0.02%** | **79.73% $\pm$ 0.05%** | **98.21% $\pm$ 0.07%** | **90.21% $\pm$ 0.08%** |
>
> | Backbone |  Method | Interpretability |   Caltech101  |    CIFAR100   |      DTD      |   Flowers102  |      PET      |
> |:--------:|:--------:|:--------:|:-------------:|:-------------:|:-------------:|:-------------:|:-------------:|
> | ViT-B-16 | CLIP | | 97.77% $\pm$ 0.12% | 86.13% $\pm$ 0.13% | 81.65% $\pm$ 0.09% | 99.10% $\pm$ 0.07% | 92.26% $\pm$ 0.14% |
> |ViT-B-16|  SpLiCE | &check; | 39.50% $\pm$ 0.23% | 71.03% $\pm$ 0.04% | 64.66% $\pm$ 0.06% | 33.99% $\pm$ 1.48% | 65.40% $\pm$ 0.14% |
> |ViT-B-16|  DN-CBM | &check; | **97.74% $\pm$ 0.12%** | 86.53% $\pm$ 0.09% | 81.19% $\pm$ 0.08% | 98.82% $\pm$ 0.07% | 90.87% $\pm$ 0.03% |
> |ViT-B-16| CLIP-DMD | &check; | 97.69% $\pm$ 0.05% | **86.58% $\pm$ 0.07%** | **81.56% $\pm$ 0.11%** | **98.90% $\pm$ 0.00%** | **92.82% $\pm$ 0.02%** |
>
> **The results show that CLIP-DMD achieves performance comparable to CLIP, a non-interpretable method, across various models and datasets**. In contrast, DN-CBM exhibits performance drops in some settings (DTD, Flowers102 on ResNet50, and all models on PET), and SpLiCE fails to match the performance of either the CLIP or CLIP-DMD.
>
> ## Weakness 3:
> **Our CLIP-DMD framework is designed to provide deeper insight into CLIP similarity.** By decomposing the similarity into its underlying concept components, it explains how the image–caption similarity is formed, rather than simply reporting a single score. Moreover, the learned sparse and interpretable features can be effectively applied to downstream tasks such as linear probing and zero-shot classification (as what we have shown in the responses for two questions above), achieving applicable performance.

---

### Comment · Area_Chair_2dCS · 2025-11-25
**Please engage with the reviewers during the rebuttal/discussion phase (if the authors plan to)**

Dear authors,

It has been over one week since the review comments were released.

If you plan to join the rebuttal, please engage sooner rather than later, so that the reviewers can have time engage as well.

Thanks,
AC

---

### Meta-Review · Area_Chair_qw6X · 2026-01-03

**Summary:**

The reviewers unanimously identified a set of critical concerns that informed the decision for this paper:

* The results relied on a single LLM for interpretability scoring, raising significant concerns about model bias.

* The zero-shot and linear probing evaluations were limited to a very small set of datasets, making it difficult to assess how general the learned sparse/interpretable features truly are.

* Reviewers noted a lack of human validation and expressed skepticism over whether the "shared concepts" actually reflect CLIP’s internal logic or are simply plausible-sounding labels.

* Figures and tables were cited as confusing, and the lack of statistical significance testing made the "human-understandable" claims feel rather anecdotal.

**Reviewer Concerns:**

Concerns Addressed by the Rebuttal

* Multi-model evaluation (`sH98`, `7TfT`, `CQqb`): The authors successfully expanded their automated evaluation to include an ensemble of VLMs (Llama-3.2, Gemma-3, and Qwen2.5). This reduced the potential for single-model bias.

* Benchmarking (`sH98`, `CQqb`): The authors provided additional results on five new datasets (Caltech101, CIFAR100, DTD, Flowers102, PET), demonstrating that CLIP-DMD outperforms SpLiCE in a supervised-feature setting.

* Clarification of Fig 1 (`PbAs`, `CQqb`): The authors provided a detailed walkthrough of their motivating figure, explaining how "sea coast" acts as a semantic differentiator.

Outstanding Concerns

* Weak human alignment (`7TfT`): While the authors provided human validation data, the alignment rates (0.59 to 0.67) are notably low. For a system intended to produce "human-understandable" explanations, having the model and humans disagree on the "best" explanation ~40% of the time suggests that the interpretability is not yet grounded.

* Restricted vocabulary (`7TfT`): The reliance on a fixed unigram/bigram corpus remains a significant bottleneck. This prevents the model from capturing complex compositional or abstract concepts that drive modern CLIP similarity.

* Statistical rigor (`PbAs`): Despite the rebuttal, the work still leans heavily on qualitative examples. The burden of proof for an interpretability paper at ICLR requires a more rigorous statistical framework beyond simple mean scores on sampled cases.

* Bias propagation (`CQqb`): The concern that the SAE might simply be encoding and amplifying existing CLIP biases—rather than "interpreting" them—remains a fundamental theoretical question that the rebuttal did not fully resolve.

**Reviewer Scores:**

* Reviewer `sH98` (Initial: 4 $\rightarrow$ Estimated: 5): Likely would have acknowledged the significant empirical effort in adding datasets and multi-VLM evals, but the "downstream application" request was only partially fulfilled.

* Reviewer `7TfT` (Initial: 4 $\rightarrow$ Estimated: 4): The mediocre human alignment scores (0.59) and the continued reliance on a restricted unigram corpus would likely have kept this reviewer below the threshold.

* Reviewer `CQqb` (Initial: 4 $\rightarrow$ Estimated: 5): The improved clarity on architecture and baseline comparisons (Grad-ECLIP) addressed their "Soundness" concerns, but the "significance of contribution" remains not fully resolved.

* Reviewer `PbAs` (Initial: 4 $\rightarrow$ Estimated: 4): Since the requested "complete rewrite" and rigorous statistical testing were not fully realized, this reviewer would likely remain critical of the quality of presentation and the "cherry-picked" nature of the evidence.

---

### Decision · Program_Chairs · 2026-01-26

Reject